# Control of tissue flows and embryo geometry in avian gastrulation

Guillermo Serrano Nájera [1,4] ✉, Alex M. Plum [2,4], Ben Steventon [1], Cornelis J. Weijer [3] ✉ & Mattia Serra [2] ✉

Embryonic tissues undergo coordinated flows during avian gastrulation to establish the body plan. Here, we elucidate how the interplay between embryonic and extraembryonic tissues affects the chick embryo's size and shape. These two distinct geometric changes are each associated with dynamic curves across which trajectories separate (kinematic repellers). Through physical modeling and experimental manipulations of both embryonic and extraembryonic tissues, we selectively eliminate either or both repellers in model and experiments, revealing their mechanistic origins. We find that embryo size is affected by the competition between extraembryonic epiboly and embryonic myosin-driven contraction—which persists when mesoderm induction is blocked. Instead, the characteristic shape change from circular to pear-shaped arises from myosin-driven cell intercalations in the mesendoderm, irrespective of epiboly. These findings elucidate modular mechanisms controlling avian gastrulation flows and provide a mechanistic basis for the independent control of embryo size and shape during development.

Morphogenesis requires spatiotemporal coordination of hundreds of thousands of cells to sculpt a developing embryo[1,2]. Elucidating the principles of morphogenesis, spanning mechanisms from molecular to organismal scales, is a grand challenge for developmental biology and the physics of living systems[3–7]. Tackling this challenge requires experimental advances and theoretical progress on two fronts: *i)* developing predictive biophysical models to test hypotheses[8–12] and *ii)* devising mathematical methods to extract key insights from experimental data[13–17]. Progress in *i* and *ii* have largely remained independent. Here, we combine mathematical methods characterizing the robust features of cumulative tissue deformation, biophysical modeling and experimental perturbations to elucidate how the avian embryo controls its dynamic geometry during gastrulation.

Recent imaging advances enable tracking single-cell trajectories[18,19], or, at the tissue level, spatiotemporal velocities describing the collective motion of cells[20–22]. However, analyzing noisy cell motion data over evolving domains can be overwhelming rather than informative. To this end, it is useful to identify robust kinematic structures. For example, given a measured ocean surface velocity field, one might be interested in understanding where floating objects accumulate or separate without knowing the precise forces generating the flows. This challenge led to the development of Lagrangian Coherent Structures[23,24], now widely adopted in science and engineering[25–27]. These are based on Lagrangian trajectories (e.g., cumulative tissue deformations) as opposed to Eulerian (or instantaneous) velocity fields, strain rates etc. (Supplementary Fig. S1 and S2). Given a velocity field over a time interval, Lagrangian Coherent Structures are dynamic curves (in 2D) and surfaces (in 3D) that organize complex trajectories. In morphogenesis, the Dynamic Morphoskeleton (DM) identifies attracting and repelling Lagragian Coherent Structures[28] (SI Sec. S1), revealing where cells maximally converge or separate over a developmental time interval of interest—dynamic 'attractors' and 'repellers' (Supplementary Fig. S1). These are attractors and repellers in the dynamical systems sense, not to be confused with causative factors like chemoattractants. The DM in chick[28–30], fruit fly[28], and zebrafish[18] embryos identified early footprints of known morphogenetic features and new ones. Repellers, in

[1]Department of Genetics, University of Cambridge, Cambridge, UK. [2]Department of Physics, University of California, San Diego, CA, USA. [3]Division of Molec. Cell and Dev. Biology, School of Life Sciences, Univ. of Dundee, Dundee, UK. [4]These authors contributed equally: Guillermo Serrano Nájera, Alex M. Plum. ✉e-mail: guillermo.serrano-najera@gen.cam.ac.uk; c.j.weijer@dundee.ac.uk; mserra@ucsd.edu

particular, were previously undocumented despite their potential relevance in dynamic tissue patterning[18,31,32].

We focus on avian gastrulation, a crucial process in early development during which a flat sheet of approximately 60,000 cells breaks symmetry, setting the vertebrate body axes and forming the three germ layers (endoderm, mesoderm, and ectoderm). At the onset of gastrulation ($t_0$, stage HH1), avian embryos consist of a circular monolayer of suspended epithelial cells, the embryo proper (EP), surrounded by an annulus of extraembryonic tissue (EE)[33–35] (Fig. 1A). Initially, mesendoderm precursor cells are in a sickle-shaped region in the posterior EP. Over the next 15 h, these mesendodermal progenitors converge and extend along the midline, undergoing individual cell ingression and forming the primitive streak (PS). This convergent extension is driven by active intercalation, apical constriction, and ingression[36–39] (Fig. 1A). These processes generate complex, embryo-scale coordinated cell movements and macroscopic vortical flows in the EP. Simultaneously, the EE expands outward (epiboly), powered by the active edge cell crawling on the vitelline membrane (Fig. 1A). The DM provides a concise summary of the complex cell movements during avian gastrulation: one line attractor and two repellers (Fig. 1B,[28]). The Attractor marks the PS, and its domain of attraction identifies the sickle-shaped region of mesendoderm cells that will ingress into the PS. Repeller 1 (R1) identifies the boundary where EP and EE regions separate. Repeller 2 (R2), bisecting the domain of attraction, identifies where anterior and posterior mesendoderm cells separate. See Supplementary Fig. S2 for Eulerian fields such as velocities and strain rates and their connection with the DM and other Lagrangian quantities.

We previously developed a minimal 2D continuum model coupling active stresses and tissue motion, finding that avian gastrulation flows arise from a mechanosensitive active stress instability, confirmed by experiments[30]. This model demonstrated that changing the initial mesendoderm pattern and a parameter associated with active ingression can reshape the Attractor. Model predictions matched experiments, showing that the Attractor of chick gastrulation flows can be altered to recapitulate other vertebrate gastrulation modes, i.e.,

their tissue flows, attractor geometries and internalization mechanisms[29,30,40]. This study, instead, investigates the origin of dynamic EP geometry (i.e., the EP-EE boundary)—specifically how the interplay between EP and EE tissues affects embryo size and the long-recognized embryo shape change from circular to pear-shaped[35,41], missing a mechanistic explanation in existing models[30,42,43]. Our previous model[30] cannot address questions of size or shape because it enforces a fixed circular EP domain, also limiting prediction and mechanistic investigation of the two associated repellers.

In this work, our model accounts for the motion and distinct myosin dynamics of both EP and EE regions, and allows for the creation and destruction of actomyosin cables. These new features predict force distributions at the EP-EE boundary, resulting in the observed size regulation of the EP region and its characteristic shape change. Combining modeling and novel experiments to symmetrically block epiboly, we find that EP size depends on the balance of myosin-driven apical constriction in the EP and pulling by EE epiboly. Surprisingly, EP size control persists when mesoderm induction is blocked. Embryo shape change, instead, arises from active intercalation driven by myosin cables in the mesoderm—even without epiboly. Our results show that chick embryo size and shape can be independently controlled. These modular tissue-scale mechanisms complement genetic modularity[44,45] and the modularity of specific cell behaviors[46], helping to rationalize how distinct embryo features emerge from coordinated cell behaviors.

## Mathematical Model

During avian gastrulation, embryonic cells frequently exchange neighbors, effectively behaving as a viscoelastic or viscous, compressible fluid deformed by active forces[30,42,43,47]. Actomyosin cables spanning 6-8 cells generate active stresses[39], driving directed cell intercalations[36–38]. We previously quantified the intensity and orientation of myosin activity[30] and modeled actomyosin cables as contractile nematic elements. In[30], myosin activity generates active stresses that drive tissue flows, altering active stress patterns. This continuum model is computationally fast, interpretable, and uses fewer

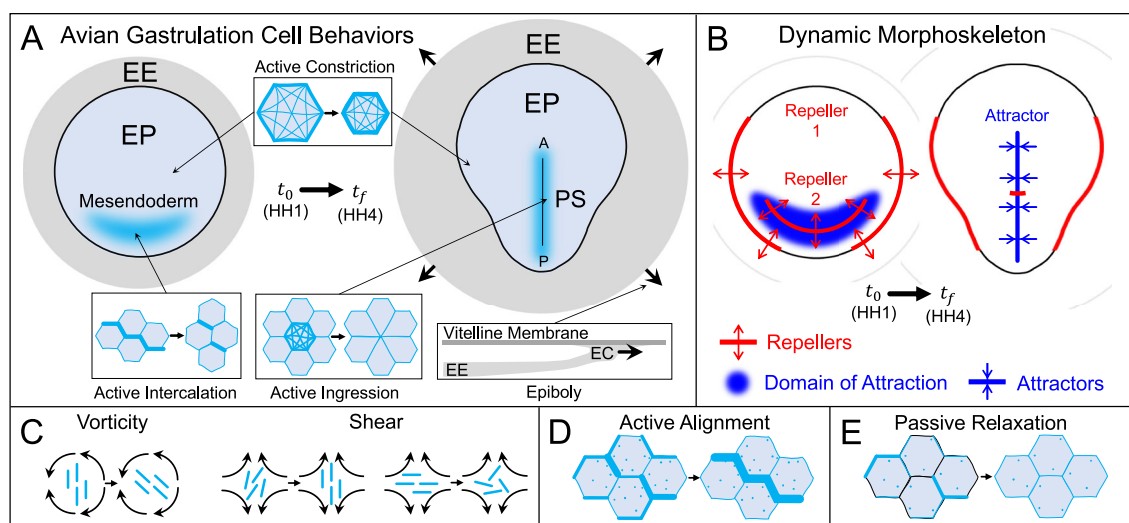

**Fig. 1 | Avian gastrulation. A** Avian morphogenesis involves convergent extension of the presumptive mesendoderm to form the primitive streak (PS), shape change of the embryo proper (EP), and extraembryonic (EE) expansion driven by edge cells (EC) crawling on the vitelline membrane. Myosin activity drives isotropic apical constriction throughout the EP and directs mesendoderm intercalations. Mesendoderm cells actively ingress at the PS, marking the anterior-posterior (A-P) axis. Cyan in insets marks elevated myosin activity. **B** The Dynamic Morphoskeleton of avian gastrulation. Repellers (R1 and R2) mark initial ($t_0$) locations of cells that maximally separate by final time $t_f$. The Attractor marks final ($t_f$) locations where

cells maximally converge during [$t_0$, $t_f$]. The domain of attraction marks the initial cell positions that converge to the Attractor. R1 lies on the EP-EE boundary, R2 bisects the presumptive mesendoderm, and the Attractor marks the PS. **C–E** Active nematic tensor dynamics (Eq. (1c)). **C** Flow coupling: Actomyosin cables (cyan nematic elements) reorient and align passively with the local flow (black). Vorticity rotates (left), while shear can rotate and both increase (middle) or decrease (right) nematic order. **D** Actomyosin cables align active junctions through mechanosensitive myosin redistribution, increasing nematic order. **E** Nematic order passively relaxes without activity or flow, decreasing nematic order.

parameters than more microscopic approaches[48,49]. Using fixed parameters, the model predicts EP gastrulation flows, specifically their attractors in wild-type (WT) and perturbed embryos[30].

This previous model, however, has limitations. First, it enforces a fixed circular EP and no EE tissue. These preclude predictions of embryo shape change[35] and repellers, which are prominent in avian morphogenesis (Fig. 1B). Second, it precludes modeling relevant distinct EP-EE dynamics. Third, it assumes that all cables are perfectly aligned with the local average orientation $\phi$. Here, we explicitly account for the EE tissue, which immunostaining shows to be largely devoid of myosin activity[30]. To account for the differing degrees of cable alignment (i.e., the presence or absence of aligned actomyosin cables), we model a nematic order parameter s, which modulates anisotropic active stress $\propto m\mathbf{Q}$, where $\mathbf{Q} = s/2[\cos 2\phi, \sin 2\phi; \sin 2\phi, -\cos 2\phi]$ is the nematic tensor and $m$ is the local fraction of available myosin generating active stress (Sec. S2.1). The $s$ dynamics, absent in the previous model, depend on flow coupling, active cable formation, and passive relaxation, enabling the creation, evolution and destruction of actomyosin cables (Fig. 1C–E, Sec. S2.2), as observed in experiments. Eq. (1) summarizes our model:

$$
\begin{cases}
\text{Force balance :} \\
\mathbf{0} = p_1[\ \overbrace{\underbrace{\nabla m}_{\text{isotropic}} + \underbrace{\nabla \cdot (m\mathbf{Q})}_{\text{anisotropic}}}^{\text{active}}\ ] + \overbrace{p_2\Delta\mathbf{v}}^{\text{viscous}} + \underbrace{\nabla[\nabla \cdot \mathbf{v}]}_{\text{dilation}} \quad (1a) \\[2mm]
\text{Dynamics of active stress intensity :} \\
\frac{\partial m}{\partial t} = -\overbrace{\mathbf{v} \cdot \nabla m}^{\text{advection}} + \overbrace{p_3(\underbrace{1-m}_{\text{activation}} - \underbrace{p_4 e^{-p_5 m}m}_{\text{deactivation}})}^{\text{cell-autonomous dynamics}} + \overbrace{p_6\Delta_{\mathbf{Q}}m}^{\text{tension propagation}} \quad (1b) \\[2mm]
\text{Dynamics of active stress orientation and anisotropy :} \\
\frac{\partial \mathbf{Q}}{\partial t} = -\overbrace{\mathbf{v} \cdot \nabla\mathbf{Q}}^{\text{advection}} + \overbrace{\underbrace{s\mathbf{D}_d}_{\text{shear}} + \underbrace{\mathbf{WQ} - \mathbf{QW}}_{\text{rigid rotation}}}^{\text{flow coupling}} + \overbrace{p_7 m\mathbf{Q}}^{\text{active alignment}} - \overbrace{p_7 s^2\mathbf{Q}}^{\text{passive relaxation}}, \quad (1c)
\end{cases}
$$

where $\mathbf{D}_d$ and $\mathbf{W}$ are the deviatoric strain rate and vorticity tensors, and $p_i$ are non-dimensional parameters (Sec. S2.5). In Eq. (1a), $p_1$ is the ratio of active stress to the characteristic viscous bulk stress and $p_2$ is the ratio of shear to bulk tissue viscosities. Eq. (1b) describes the dynamics of $m$, the fraction of locally available myosin generating active stresses in (1a) (Sec. S2.1). $p_3$ is the ratio of gastrulation's characteristic timescale ($t_c \approx 15\,$h) to the cumulative timescale for converting all available myosin into tissue-scale active stress (Sec. S2.1). $p_4$ is the ratio of myosin tissue-scale activation to deactivation timescales. $p_5$ is the mechanosensitivity of the deactivation rate to cable tension[30], which results in an active stress instability and reflects the exponential decrease observed in single-molecule experiments[50,51]. This active stress instability ensures the divergent behavior of EP and EE regions, consistent with experiments (Fig. 2E, Sec. S2.1, Fig. S3A and Fig. S9). $p_6$ is the ratio of mechanical (or tension-induced) propagation of myosin activity along the cables to transport via advection[30]. The first three terms in Eq. (1c), standard in active nematics[52–54], account for transport and flow-induced effects on $\mathbf{Q}$ (Fig. 1C). Notably, with only flow coupling and advection, convergent extension invariably destroys initial cable alignment, driving $s$ to 0 in the vicinity of the streak (Fig. S3C, Sec. S2.2), inconsistent with experiments (Fig. 1 of[30]) and reflecting the loss of aligned high-myosin junctions from executing sequential cable contractions[55,56]. To sustain convergent extension, an alignment mechanism is needed to counteract order destruction from self-induced flows (Fig. S3D). The active alignment term ($p_7 m\mathbf{Q}$) models actomyosin cable formation as an active process, conserving myosin but polarizing active stress to represent redistribution of myosin to junctions already under higher tension due to myosin activity[38,39,57,58] (Fig. 1D). This agrees with experimental observations that myosin cables favor asymmetric stress propagation and that their formation can be inhibited by myosin inhibitors, as well as the

hypothesis that they acquire their super-cellular alignment through a self-organizing stress-dependent mechanism[38]. It also reflects in the continuum setting a mechanism recently explored in vertex models, with local tension feedback redistributing myosin to increase anisotropy while conserving $m$[49,56,59]. By contrast, passive relaxation ($-p_7 s^2\mathbf{Q}$) reduces $s$ accounting for cell shape rigidity and random intercalations[56,59,60] which would bring the tissue to an isotropic state (Fig. 1E). $p_7$ is $t_c$ times the activity-induced alignment and passive relaxation rate (Sec. S2.5). Active alignment and passive relaxation terms, introduced in active nematics[52,61], are yet unexplored in the context of morphogenesis.

As boundary conditions in Eq. (1a), we prescribe an outward normal velocity $v_e$ consistent with observed migrating edge cells (Sec. S2.6, Fig. S4[62]). For Eqs. (1b-1c) we impose no flux for $m$ and $\mathbf{Q}$. For the initial condition of Eq. (1b), we use immunostaining data indicating minimal myosin activity in the interior EE throughout gastrulation, while myosin activity increases over time in the EP (Figs. 1E, S10 in[30] and Fig. S9), consistent with an $m$ instability (Fig. S3A, Sec. S2.1). To account for the distinct myosin dynamics of EP-EE cells, we set $m(\mathbf{x}_{EP}, t_0) > m^*$ and $m(\mathbf{x}_{EE}, t_0) < m^*$ where $m^*$ is the unstable uniform fixed point in the myosin dynamics (1b) (Supplementary Fig. S3A). Additionally, we initialize higher myosin activity in the posterior EP, matching the sickle shape of mesendoderm precursors[29,30] (cf. scalar field in Fig. 2A). For the initial condition of Eq. (1c), we set $\phi(\mathbf{x}, t_0)$ along the tangential direction, matching experimental observations[30], and determine the initial nematic order from $m(\mathbf{x}, t_0)$:$s(\mathbf{x}, t_0) = \sqrt{m(\mathbf{x}, t_0)}$, the equilibrium of Eq. (1c) before the onset of flows (Sec. S2.2). This reflects the expectation that a mild tangential cable alignment is amplified by active contraction and myosin redistribution in the mesendoderm precursor region (Fig. 2A) in line with experiments[30,38]. For details on boundary and initial conditions and the numerical scheme, see Sec. S2.6–S2.8.

## Results

### Model predicts repellers and embryo shape change

Solving Eq. (1), with the above initial and boundary conditions, we recapitulate avian gastrulation flows, including their characteristic Eulerian features: dynamic active stress patterns (Fig. 2A−C), vortices, and convergent-extension flows (Fig. 2D); and Lagrangian features: R1, R2, and EP shape change (Fig. 2F, H). See Supplementary Movie 1 for the time evolution of these fields and Fig. S6 for shape quantification. Figure 2B and C show increased EP myosin activity ($m(\mathbf{x}, t_f)$) and increased anisotropic activity along cables ($m(\mathbf{x}, t_f)\mathbf{Q}(\mathbf{x}, t_f)$), consistent with experiments[30]. Figure 2D shows velocity and velocity divergence at final time $t_f$ = stage HH4 consistent with observations of sustained vortical flows, negative divergence in the PS (Attractor), lower divergence magnitude in the rest of the EP and positive divergence in the EE region[30,38] (Fig. S7). Immunostaining of HH4 embryos confirms high active myosin in the EP and low in the EE (Fig. 2E), consistent with our previous work[30].

To assess Lagrangian features, Fig. 2F shows an initially uniform grid and circular EP-EE boundary (Fig. 2A) advected with the model velocity ($\mathbf{v}$), showing lower expansion of the EP relative to the substantial stretching of the EE[55]. With the EP-EE boundary now free to deform, the model also successfully predicts the EP's long-recognized geometric transformation from circular to pear-shaped[35,63] (also plotted in Fig. 2B–D, F). This nontrivial aspect requires the correct self-organizing dynamics of active forces (Movie 1 and Sec. S3.1) on the EP boundary, internal points of our modeling domain. Figure 2H shows the repellers in the highest Lagrangian stretching field $_2\lambda_{t_0}^{t_f}(\mathbf{x}_0)$, displayed at the embryo's initial configuration[28]. Throughout, $_2\lambda_{t_0}^{t_f}(\mathbf{x}_0)$ denotes the largest singular value field of the deformation gradient tensor $\nabla_{\mathbf{x}_0}\mathbf{F}_{t_0}^{t_f}(\mathbf{x}_0)$, where $\mathbf{F}_{t_0}^{t_f}(\mathbf{x}_0)$ denotes trajectories from $t_0$ to $t_f$ starting at $\mathbf{x}_0$ (Sec. S1). High values of $_2\lambda_{t_0}^{t_f}(\mathbf{x}_0)$ mark initial embryo locations (repellers R1 and R2) where nearby cells maximally separate

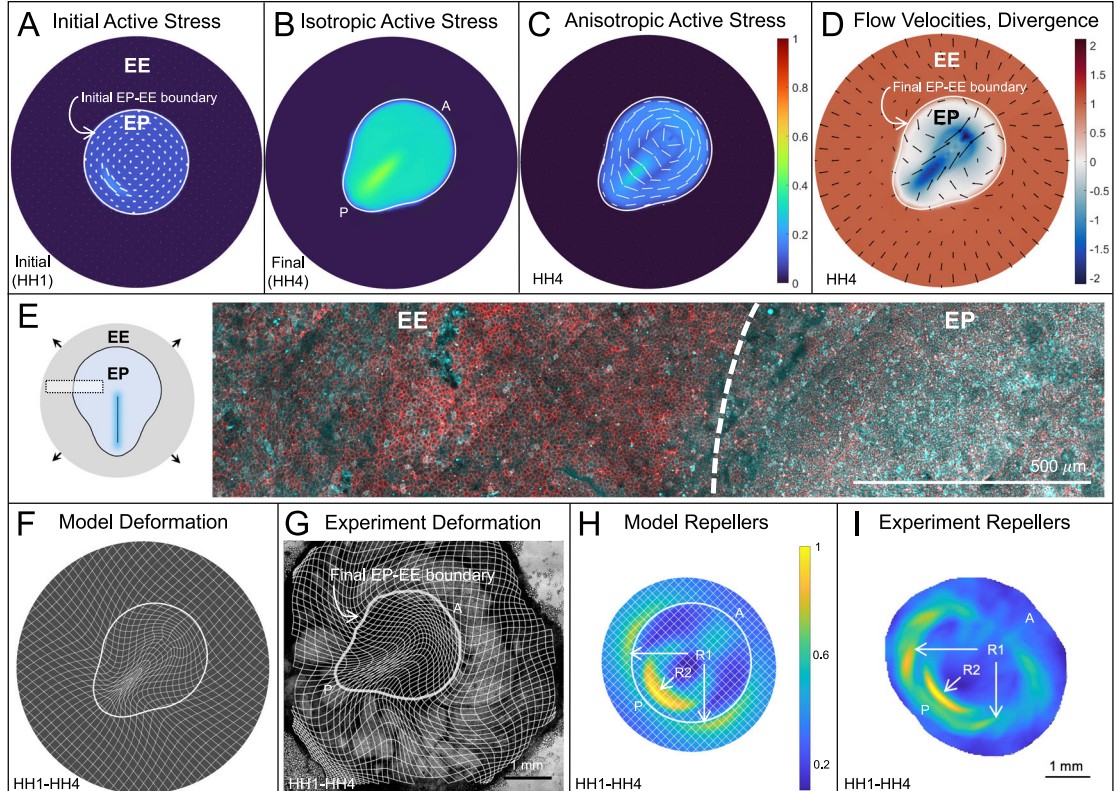

**Fig. 2 | Wild-type gastrulation. A** Initial distribution of myosin activity $m$ with orientation $\phi$ (white director field). Director lengths increase with $s$ (exponentially to ease visualization). White circle marks the initial EP-EE boundary, updated at later times (**B–D**, **F**) by advection with the model **v**. **B**, **C** Final time ($t_f$) distribution of isotropic (**B**, color bar $m$) and anisotropic (**C**, color bar $ms$ with directors as in **A**) myosin activity. **A–C** share the color bar. **D** Predicted model velocity **v** and divergence $\nabla \cdot \mathbf{v}$ at the final time ($t_f$). All model quantities are dimensionless. **E** Confocal image of a representative section of an HH4 stage WT chick embryo at the position indicated on the left. Staining for actin (phalloidin, red) and doubly phosphorylated myosin light chain (cyan) in the embryo. **F**, **G** Deformed Lagrangian grids and final EP-EE boundary in model (**F**) and experiment (**G**). **G** Experimental grid overlaid on a bright-field image of an HH4 embryo, with EP-EE boundary located by drawing an initial polygon around R1 (Sec. S5.4) and advecting it with experimental **v**. **H**, **I** Repellers (R1 and R2) in model (**H**) and experiment (**I**), marked by ridges of $_2\lambda_{t_0}^{t_f}(\mathbf{x}_0)$ normalized by the spatial maximum, and displayed on the initial undeformed configuration (Sec. S1). A, P labels mark the anterior-posterior axis. See Supplementary Movie 1 for the WT model velocity, velocity divergence, isotropic and anisotropic stresses, active forces, repellers, attractors, and Lagrangian grids over time. Movie repellers and attractors colorbars are normalized over their spatio-temporal maxima, achieved at $t_f = HH4$.

by $t_f$ (Fig. S1). To visualize these repellers (constituting the DM), we normalize $_2\lambda_{t_0}^{t_f}(\mathbf{x}_0)$ by its maximum. The model reproduces the circular arcs of R1 at the EP-EE boundary and R2 in the posterior[28] (c.f. Fig. 1B). Altogether, these results showcase the model's predictive capability, providing only the initial myosin intensity ($m(\mathbf{x}, t_0)$), cables distribution ($\mathbf{Q}(\mathbf{x}, t_0)$), and epiboly velocity as inputs.

To test our model and explore perturbations, we developed an imaging and analysis pipeline using bright-field microscopy, enabling us to simultaneously track tissue flows in up to ~16 chick embryos without fluorescent labels (Fig. S5). This approach demonstrates the robustness of the DM across several embryos, despite the intrinsic variability of their tissue flows (Fig. S8). Representative Lagrangian deformation (Fig. 2G) and repellers (Fig. 2I) from this pipeline are consistent with model predictions (Fig. 2F, H). Importantly, our goal is not to precisely match the DM between the model and a specific experiment by fitting parameters—this would depend on specific experiments. Instead, we aim to predict the existence and average geometry of repellers, which are robust across experiments (Fig. S8).

### Origin and elimination of Repeller 1

We next investigate the mechanisms underlying R1. Edge cells adhering to the vitelline membrane crawl outwards, pulling the EE and EP radially (Fig. 1A). Edge cells crawling rates are nearly constant and robust to several treatments, including excision of the EP[64–66],

justifying our boundary condition for the velocities in the model (Sec. S2.6). Epiboly contributes to global tension[62,67,68] that propagates to the EP, as severing the EP-EE boundary causes both regions to contract[68]. Yet, R1 shows that EP and EE cells separate, implying distinct expansion rates. Tension can arise from passive stretching and active contractility. We hypothesize that the EP resists epiboly-driven expansion with isotropic myosin activity (Figs. 1A, 2A, B, E), which constricts cells' cortical cytoskeletons (active contractility), allowing them to bear the isotropic tension contributed by epiboly without stretching like EE cells. Isotropic myosin activity can constrict cells via circumferential purse-string contraction[69] or medial apical network contraction[70,71] (Fig. 1A). Before the onset of motion, apical areas in the EP are already slightly smaller in the embryo and smallest in the posterior[38,72], consistent with elevated myosin activity. As gastrulation proceeds, EP tissue thickens, and apical cell areas decrease[62,72]. Decreasing EP cell areas accompany the marked rise in EP myosin activity, with the greatest shrinkage near the PS where myosin activity grows highest[38,72]. In contrast, EE cells, devoid of active myosin (Fig. 2E), stretch thin, and their cell areas can increase more than double[72,73]. These distinct EP-EE dynamics are consistent with the model's active stress patterns and cumulative deformation (Fig. 2A, B, F).

To test our hypothesis that R1 results from the opposition between EP inward constriction and edge cell outward crawling, we first eliminated epiboly in the model ($v_e = 0$), solving Eq. (1) with the

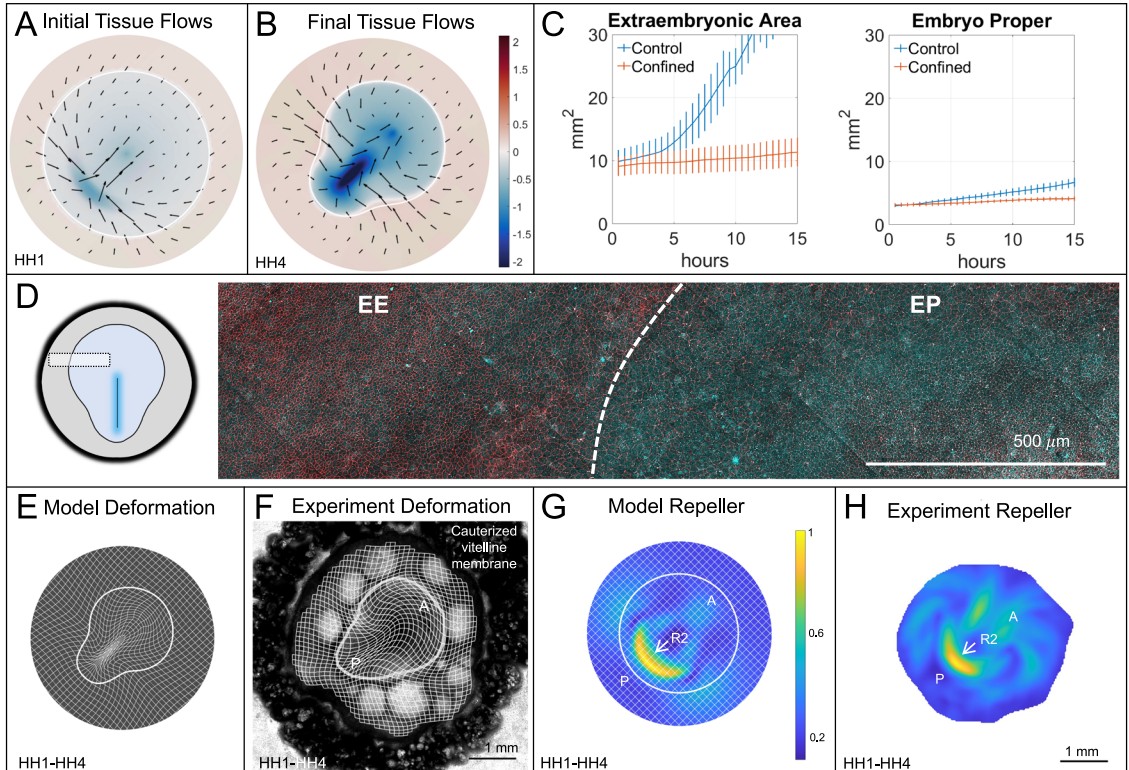

**Fig. 3 | Repeller 1 elimination.** Confining the embryo eliminates R1 but preserves R2. (**A, B**) Predicted model velocity **v** and divergence $\nabla \cdot \mathbf{v}$ at $t_0$ (**A**) and $t_f$ (**B**), with $v_e = 0$ reflecting embryo confinement. **C** Mean and standard deviation of experimental EP and EE areas over 15 h ($N = 8$ per group). **D** Doubly phosphorylated myosin light chain (cyan) and actin (phalloidin, red) in a representative section of a confined embryo. **E, F** Deformed Lagrangian grids and final EP-EE boundary in model (**E**) and experiment (**F**). Experimental Lagrangian grid overlaid on the bright-field image of a confined HH4 chick embryo, boundary located as in Fig. 2. Embryo confined in experiments by cauterizing the EP side of the vitelline membrane. **G, H** Repeller (R2) in model (**G**) and experiment (**H**), marked by ridge of $_2\lambda_{t_0}^{t_f}(\mathbf{x}_0)$, normalized by the spatial maximum, and displayed on the initial undeformed configuration. Compare (**G, H**) with the corresponding WT Fig. 2H, I. A, P labels mark the anterior-posterior axis. Supplementary Movie 2 shows the time evolution of Lagrangian and Eulerian model fields as in Supplementary Movie 1.

same parameters, initial and boundary conditions as in Fig. 2. Eliminating epiboly dramatically weakens differential EP-EE expansion and effectively eliminates R1 relative to R2 and EP shape change, which are not substantially altered (Figs. 3A, B, E, G). Eliminating isotropic myosin activity or the distinct initial distribution of myosin between the EP and EE ($m(\mathbf{x}, t_0)$) is also sufficient to eliminate R1 in the model (Sec. S3), but these perturbations are experimentally infeasible. Additionally, eliminating epiboly may affect the average isotropic tension of the tissue[62], reducing the rate of myosin accumulation, but any such effect is beyond the scope of our current model. To experimentally verify our prediction, we restricted epiboly movements. Past approaches to interfering with epiboly have limitations: ablating edge cells blocks epiboly only transiently as new edge cells quickly differentiate and resume epiboly[67]; removing the EE entirely can compromise development because of its signaling role patterning the early EP[68,74]; chemical treatments like colchicine also disrupt cell division[75,76]; and manually wrinkling the vitelline membrane[68] or extracting yolk[77] to reduce vitelline membrane tension only partially reduce expansion and not always uniformly. To overcome these limitations, we developed a novel technique, cauterizing the inner face of the vitelline membrane surrounding the EE using a soldering iron (Fig. 3F and Sec. S5.2). The edge cells cannot adhere to the cauterized region, creating a fixed boundary that uniformly blocks epiboly.

This confinement technique is compatible with our high-throughput live imaging pipeline (Fig. S5). Figure 3C confirms that confinement prevents embryo expansion and that the marked difference in EE-EP expansion observed in the WT embryo (blue curves) has been reduced (orange curves). Immunostaining of phosphorylated

(active) myosin verifies persistent active contractility across the EP under confined conditions (Fig. 3D and Fig. S9). A representative Lagrangian deformation grid of the confined embryo (Fig. 3F) shows lesser differential EP-EE stretching than WT (Fig. 2G) and that the embryo still becomes pear-shaped (Fig. 3F, and Suppl. Fig. S6) and forms a PS (Fig. S11B). Figure 3H indicates the presence of R2 but no R1 (compare with Fig. 2I), consistent with model predictions (Fig. 3G). We find that the confined embryos develop well past gastrulation stages in these conditions (Fig. S10), suggesting that epiboly may be unnecessary at gastrulation stages (Discussion and Sec. S4). These results suggest that R1 arises from a tug-of-war between EE epiboly and EP active constriction. Without epiboly, no substantial separation occurs at the EP-EE boundary.

**Origin and elimination of Repeller 2**
Preserving R2 without R1 suggests that avian gastrulation's repellers may arise from independent mechanisms. We hypothesize that R2, bisecting the initial crescent-shaped mesendoderm, results from acto-myosin cable-directed intercalations in this region[38,39], independent of epiboly and EP constriction associated with R1. To test our hypothesis, we eliminated the model mesendoderm, represented by elevated posterior myosin activity and anisotropy, solving Eq. (1) with the same parameters, initial and boundary conditions as in WT. This also eliminates the initially elevated posterior cable anisotropy associated with active alignment (Fig. 2A, Sec. S2.7). With this change, the EP remains circular and retains differential EP-EE expansion (Figs. 4A, B, E). Figure 4G shows the elimination of R2, but R1 remains and becomes circularly symmetric due to the retention of epiboly and uniform isotropic active stress in the EP. Eliminating anisotropic active stress or

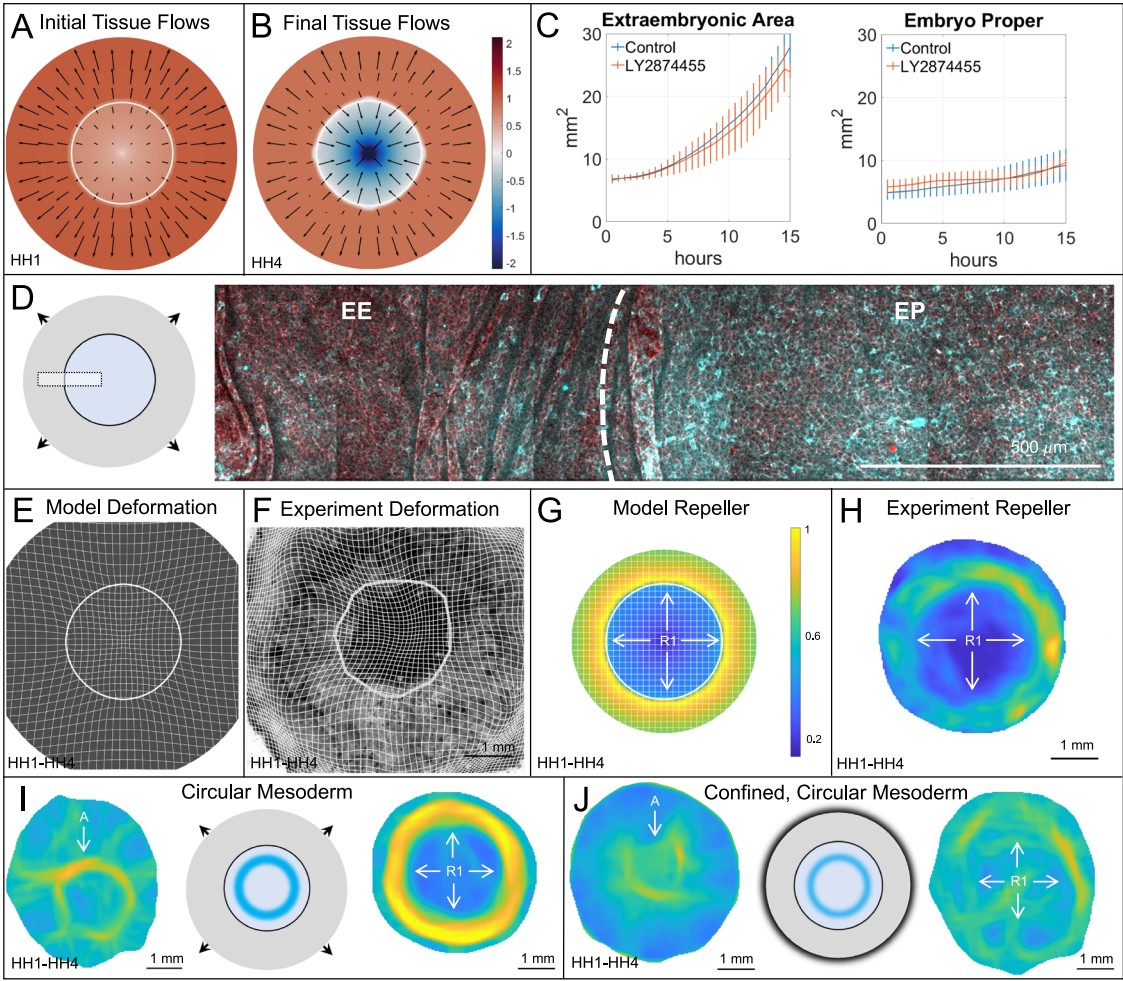

**Fig. 4 | Repeller 2 elimination.** Blocking mesendoderm induction eliminates R2 but preserves R1. **A, B** Predicted model velocity **v** and divergence $\nabla \cdot \mathbf{v}$ at $t_0$ (**A**) and $t_f$ (**B**). Model results obtained solving Eq. (1) with uniform initial EP myosin reflecting the absence of mesendoderm. **C** Mean and standard deviation experimental EP and EE areas over 15 h ($N = 4$ per group). **D** Doubly phosphorylated myosin light chain (cyan) and actin (phalloidin, red) in a representative section of a treated embryo. **E, F** Deformed Lagrangian grids and final EP-EE boundary in model (**E**) and experiment (**F**). Experimental grid overlaid on bright-field image of a treated HH4 chick embryo, boundary located as in Fig. 2. Mesendoderm induction blocked in experiments using $1\mu M$ LY2874455, a pan-FGF receptor inhibitor. **G, H** Repeller (R1)
in model (**G**) and experiment (**H**), marked by ridge of $_2\lambda_{t_0}^{t_f}(\mathbf{x}_0)$, normalized by the spatial maximum, and displayed on the initial undeformed configuration. Compare (**G, H**) with the corresponding WT Fig. 2H, I. Supplementary Movie 3 shows the time evolution of model fields as in Supplementary Movie 1. Movie attractor colorbar is normalized using the WT spatiotemporal maximum to emphasize the relative lack of deformation (see Supplementary Fig. S11 for a full comparison of different treatments). **I, J** Experimental data for circular mesoderm (FGF treatment) in unconfined (**I**) and confined (**J**) conditions. Colorbars as in **G**, showing $_2\lambda_{t_f}^{t_0}(\mathbf{x}_f)$ (attractors, left) and $_2\lambda_{t_0}^{t_f}(\mathbf{x}_0)$ (repellers, right) normalized by the spatial maximum.

active cable formation ($p_7 = 0$) is also sufficient to eliminate R2 (Sec. S3).

We previously showed that oriented cell intercalation and oriented myosin cables are characteristic properties of the mesendoderm cells[29]. Indeed, using a pan-FGF receptor inhibitor LY2874455[78] completely blocks mesendoderm formation and the oriented cell intercalation associated with PS formation[28,29]. To verify our model prediction, we experimentally blocked mesendoderm formation using the same chemical treatment (LY2874455 $1\mu M$). Fig. 4C demonstrates that this treatment leaves the EP and EE areal expansions unaltered compared to WT. Further, immunostaining of treated embryos shows that elevated phosphorylated myosin in the EP is not affected by the FGF inhibition (Fig. 4D). Consistent with model results, the embryo retains a circular geometry (Fig. 4F, Supplementary Fig. S6), keeping R1, which becomes circularly symmetric, while eliminating both R2 and the Attractor (Figs. 4H, and Supplementary Fig. S11C). These results suggest that R1 arises from distinct mechanisms, independent of those jointly generating R2 and the Attractor. They also suggest that elevated EP actomyosin activity represents an inherent property of the EP, not

requiring mesendoderm induction. Instead, induction is only required to assemble anisotropic cables perpendicular to the midline and execute the directed intercalations that contribute to forming the PS and embryo shape change.

Mesoderm is necessary for both R2 and the Attractor. These co-occur in WT chick development because the crescent-shaped mesoderm's motion into the PS requires convergent extension, contributing to both R2 and a linear attractor (Sec. S3.2). To test the role of mesoderm geometry, we used FGF to generate a circular mesoderm domain, resulting in circular myosin cables[29,79]. In the model and experiments, this circular geometry results in convergence towards a circular attractor[30], instead of an extending line attractor, reinforcing a symmetric R1 and eliminating R2 (experiments in Fig. 4I and model in Supplementary Fig. S12A). Furthermore, we show that combining FGF addition with confinement, the circular attractor persists, R2 remains absent, and circular contraction recovers a weak R1 (Figs. 4J, Supplementary Fig. S12B). This clarifies the role of mesoderm geometry in jointly generating R2 and the (line) Attractor, associated with embryo shape change.

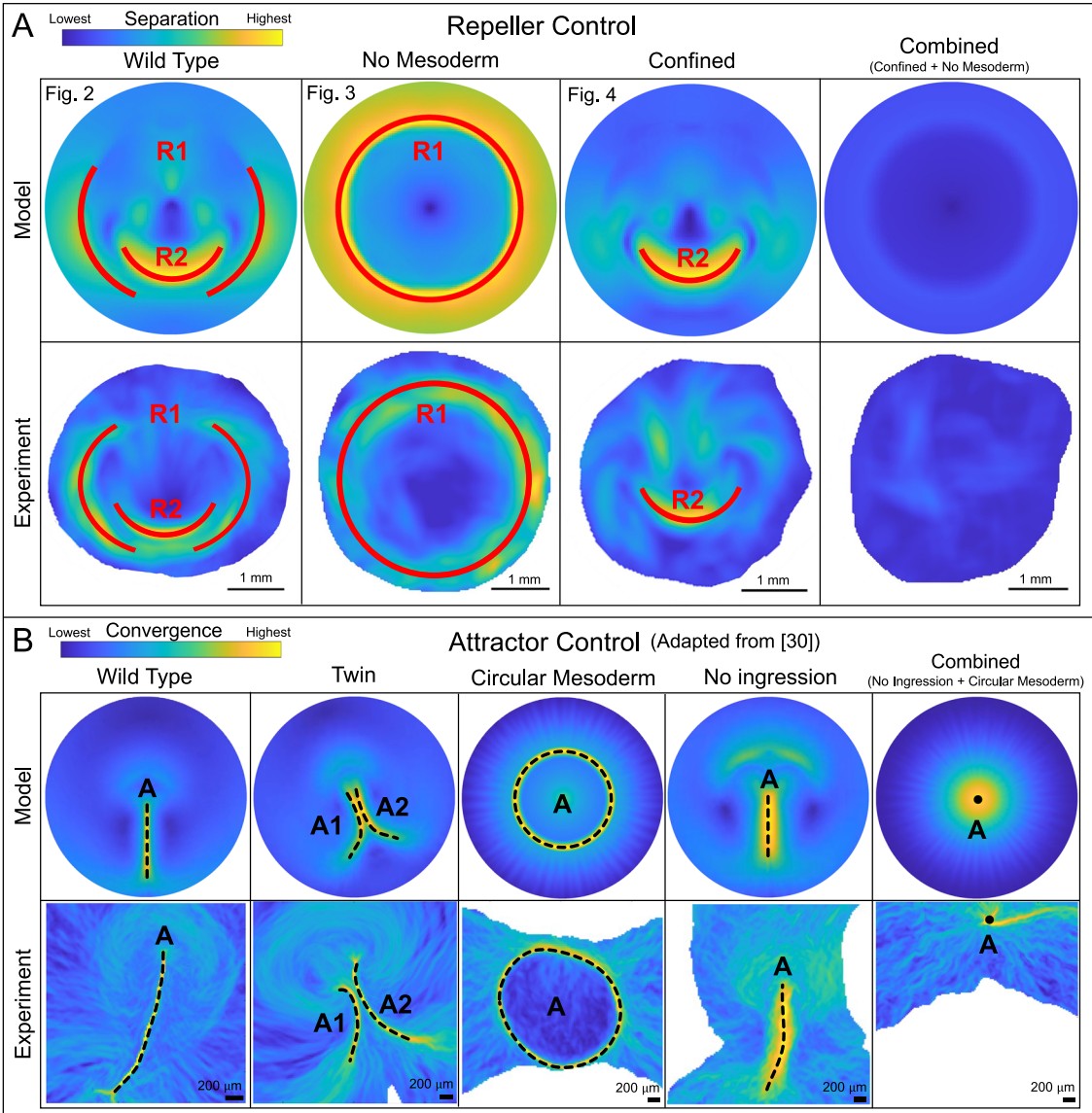

**Fig. 5 | Controlling the Dynamic Morphoskeleton: Repellers and Attractors.**
**A** Repellers R1 and R2 (red) can be controlled combinatorially by confining the chick embryo and inhibiting mesendoderm induction. First three columns adapted from Figs. 2–4, showing $_2\lambda_{t_0}^{t_f}(\mathbf{x_0})$ fields normalized by their spatial maxima. Fourth column shows perturbation simultaneously confining the embryo and inhibiting mesendoderm induction in the model and experiments with $_2\lambda_{t_0}^{t_f}(\mathbf{x_0})$ normalized by the WT spatial maximum to emphasize the lack of deformation. Supplementary

Movie 4 shows time evolution of model fields associated with the fourth column. **B** Control of attractor shapes (dashed black) in model and experiments by modulating cells' ability to ingress (16 hr treatment with 100 nM of VEGF receptor inhibitor axitinib) and initial mesendoderm shape (50 μg/ml FGF2 on the hypoblast). See[29] for experimental details. Fields show $_2\lambda_{t_f}^{t_0}(\mathbf{x_f})$, normalized by their spatial maximum. **B** adapted from previous work[30].

Finally, combining embryo confinement ($v_e = 0$) with inhibition of mesendoderm induction (uniform EP $m(\mathbf{x}, t_0)$), we simultaneously eliminated both repellers in the model and experiments (Fig. 5A, last column). Without epiboly and mesoderm-driven active intercalation, minimal deformation occurs, resulting in no shape change (Fig. S6), differential EP-EE expansion, or Attractor formation (Fig. S11D).

## Discussion

In this study, we have identified the mechanisms and cellular behaviors underlying multicellular flow repellers during avian gastrulation and how embryos change size and shape, from circular to pear-shaped (Figs. 1, 2). Combining in-vivo experiments, active matter theory and nonlinear dynamics, we find that R1, separating EE from EP, arises from the balance between outward epiboly and active constriction of EP cells (Fig. 4) and is associated with embryo size. In contrast, embryo

shape changes, PS-formation (Attractor), and R2—separating anterior and posterior mesoderm—require active intercalations by mesendoderm cells, sustained by active cable formation (Fig. 3). Furthermore, the circular symmetry of R1 in the absence of R2 (Fig. 4, and Supplementary Fig. S11) clarifies their relationship in WT (Fig. 2). With both repellers present, the EP shape change associated with R2 accentuates EP-EE separation at the lateral posterior EP edges, breaking R1's otherwise circular symmetry. In general, the relative strength ($_2\lambda_{t_0}^{t_f}$ ridges) of R1 and R2 depends on the relative timing of epiboly and mesoderm-driven convergent extension (Fig. S13). See Supplementary Table 3 and Supplementary Movies 5–28 for additional model perturbations and sensitivity analysis.

Our findings were guided by the Dynamic Morphoskeleton, which summarizes noisy spatiotemporal tissue flows in discrete kinematic units (attractors and repellers, Fig. 1B)[28], robust to

intrinsic embryo variability (Fig. S8), owing to their cumulative (Lagrangian) nature that integrates cellular trajectories over time. Previous studies have also used instantaneous (Eulerian) quantities to analyze avian gastrulation flows, such as the strain rates[29], vorticity[80,81] and the Helmholtz decomposition to decouple flow rotation and compressibility[42]. Eulerian quantities are suitable for capturing instantaneous patterns of the velocities and their persistence or changes (Figs. 3A, B, 4A, B, and Supplementary Movies 1–4). Lagrangian analysis complements these techniques by quantifying cumulative tissue deformation while losing details of instantaneous flow patterns. The independent manipulability of R1 and R2 (Fig. 5A) and their mechanisms suggest modular properties. This resonates with the notion of morphogenetic modularity: complex systems, like developing embryos, are composed of semi-autonomous modules, modifiable without disrupting the overall system's functionality[46,82]. In our recent findings, we also controlled the shape of avian gastrulation's Attractor (Fig. 1B)[29,30], modifying its wild-type line shape (PS) into a circle, dot, or thicker shorter line (Fig. 5B) by independently modulating the shape of the presumptive mesendoderm territory and the capacity of mesendoderm cells to undergo ingression. Altogether, the combinatorial elimination of repellers and alteration of attractors reveal the plasticity of the early embryo, providing insight into the evolvability of gastrulation programs[40].

R1 arises from the tug-of-war between EE epibolic movements and the intrinsic active contractility of the EP. Surprisingly, when we confined embryos to eliminate epiboly while maintaining an intact EE (Supplementary Movie 29), embryos still gastrulated and developed proportioned axial structures, despite having shorter body axes (Supplementary Fig. S10). Similarly, a recent report showed that decreasing tension in the vitelline membrane delayed epiboly, resulting in a shorter body axis[77]. Disrupting epiboly progression in zebrafish does not prevent gastrulation but also produces a shorter body axis[83,84]. These findings suggest that while epiboly is crucial for nutritive functions later in development[85], it is not required for the early stages of avian embryogenesis. The active contractility of the avian embryo may have evolved as a mechanism to resist the influence of epiboly, allowing the embryo to maintain its intrinsic (size-dependent) patterning mechanisms and developmental timeline. Indeed, experiments show that partial EP ablations normally require detachment of the EE from the vitelline membrane to avoid the embryo ripping apart[86], suggesting that epiboly forces can pull apart a mechanically compromised EP. In contrast, R2 and the Attractor jointly reflect embryo shape changes and PS formation from an initially crescent-shaped mesoderm. In chick, initially close cells starting on opposite sides of R2 experience large separation during gastrulation (for visualization, see Fig. 3A of[28]), which may aid compartmentalization along the PS[32]. The independent origins and modular properties of these repellers provide a new perspective on the robustness and evolvability of avian gastrulation (Sec. S4).

Our non-cell-autonomous model (Eq. (1)) consists of a viscous, compressible, active nematic flow driven by EE epiboly and EP-EE distinct myosin dynamics. $m$ generates isotropic active stress modeling active cell constrictions and ingressions, and anisotropic stresses generating active intercalations. Anisotropic active stresses arise from three key variables: cable orientation $\phi$, $m$ and order parameter $s$, quantifying the presence of aligned actomyosin cables, and whose dynamics involve non-standard activity-induced alignment and passive relaxation terms. These two processes capture the creation and destruction of aligned actomyosin cables, which are ubiquitous in experiments and critical for sustaining gastrulation flows. Using initial observed values of $\phi$, $m$ and scalar space- and time-independent parameters, our model accurately predicts 15 h of avian gastrulation flows and embryo shape changes. These shape changes, not explained by existing models[30,42,73], are nontrivial as they require prediction of

active force distribution on the EP boundary, internal points of our modeling domain (Supplementary Movies 1–4 and Sec. S3.1), dynamically shaping the embryo from a disc to a pear shape. Increasing shear viscosity, reflecting stiffer cell-cell junctions, results in poor EP shape change and disrupts vortical movements (Sec. S3), consistent with theoretical[87–89] and experimental[42,90,91] studies suggesting that division may contribute to tissue fluidity. Our model captures the dynamic tissue flows despite not explicitly modeling cell divisions or ingressions. Incorporation of these processes could enable predictions of cell area dynamics, clarify viscoelastic contributions, including mechanical signals from the rising tension contributed by epiboly, and connections to active solid models[11,56,59]. Likewise, explicitly modeling multiple cell types and their differentiation will be important to clarify feedback between tissue flows and concurrent dynamic tissue patterning[32].

## Methods
Our research complies with all relevant ethical regulations. We describe experimental data acquisition and analysis methods in SI Sec. 5. We describe methods for the numerical computation of the Dynamic Morphoskeletons in SI Sec. 1. We describe methods for the numerical simulation of the mathematical model in SI Sec. 2.

### Reporting summary
Further information on research design is available in the Nature Portfolio Reporting Summary linked to this article.

## Data availability
We included lightweight experimental velocity fields describing the tissue flows described in Figures 2–5, as well as the measurements for Figure S10 and statistical analysis in the Source Data file. Additional data used in this paper are available from the corresponding authors upon request. Source data are provided with this paper.

## Code availability
The code used in this paper is available from the corresponding authors upon request.

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

## Acknowledgements

We acknowledge Dillan Saunders for insightful discussions and Alexandra Neaverson, Yuri Takahashi, Apolline Delahaye, Maciej Żurowski and Sreejith Santhosh for their input and experimental support. We also acknowledge Fridtjof Brauns for suggesting Supplementary Fig. S2. GSN acknowledges support from Leverhulme Trust Early Career Fellowship (ECF-2022-474). AP acknowledges support from the National Institutes of Health (NIH) under training grant number T32-GM127235. CJW acknowledges support from the BBSRC (BB/N009789/1, BB/K00204X/1, BB/R000441/1, BB/T006781/1) and the Wellcome Trust(101468/Z/13/Z). MS acknowledges support from the Hellman Foundation, NSF PHY-2413073 and NSF CAREERPHY-2443851.

## Author contributions

G.S.N. and M.S. designed the research. G.S.N. designed, performed, and analyzed experiments, with inputs from B.S. A.P. and M.S. formulated the mathematical model. A.P. performed and analyzed numerical simulations. A.P., G.S.N., C.J.W. and M.S. wrote the manuscript. C.J.W. and M.S. supervised the project.

## Competing interests

The authors declare no competing interests.
