## [Peer Review file · Nature Communications]

Control of Tissue Flows and Embryo Geometry in Avian Gastrulation

Corresponding Author: Professor Mattia Serra

Version 0:

Reviewer comments:

Reviewer #1

(Remarks to the Author)

The paper uses a refined version of an active matter model (Serra et al. 2023) to examine tissue flow in the avian embryo. It is based on the previously developed concept of a Dynamic Morphoskeleton (DM) (Serra et al. 2020) whose fundamental units are attractors and repellers of moving cells. As shown previously, in the avian embryo Repeller 1 separates extraembryonic tissue (EE) and the embryo proper (EP), Repeller 2 and an Attractor are located at the prospective mesoderm. In the present manuscript, experiments and modeling reveal the mechanistic origins of repellers, which suggests their independence as morphogenetic modules. The new model includes the EE and adds the modulation of anisotropic stress, and it correctly predicts gastrulation movements including the development of the EE-EP boundary.

The performance of the improved model captured in Equation 1 is impressive. However, instead of enforcing the DM concept the results obtained seem to weaken it. The DM had been introduced to replace the Eulerian representation of tissue flow through reference frame-dependent cell trajectories and streamlines by a frame-invariant description that identifies attractors and repellers, i.e. tissue regions that show maximal convergence or separation of cells over a given period of time. However, the issue of frame invariance seems not so important when the whole embryo is mapped anyway as in the present paper, instead of a select subregion. And it seems that mechanistic insights into gastrulation movements like convergent extension in the PS region or EE and EP moving apart in response to epiboly are easier to gain intuitively, at the cellular level (insets in Fig.1A), when streamlines are considered.

On the other hand, the Lagrangian features extracted are kinematic units that are not directly linked to cell level mechanisms. For example, as shown here, Repeller 1 is generated by forces outside its domain, by epibolic edge-cell crawling right at the margin of the whole embryo and by general constriction in the EP – both features well represented in the Eulerian but not so obvious in the Lagrangian approach. There, the term Repeller evokes the idea of a local activity, not of a passive reaction to outside forces. One seems to be prompted to look for an active mechanism in the wrong place. Moreover, as shown in the elegant confinement experiment (which could have been conceived without being aware of Repeller 1, just from curiosity about the effect of epiboly on gastrulation), the presence of Repeller 1 is of minor relevance for gastrulation. As to Repeller 2, its origin can be traced to cell intercalation in the prospective mesoderm which also contributes to the Attractor (in addition to ingression). Thus, the same basic cellular process contributes to the Attractor and to the Repeller 2, which might be better seen when the Attractor is depicted as DOA at the same stage HH1 as Repeller 2 – but perhaps best when streamlines are considered.

A conclusion from these latter considerations is that Attractor and Repeller 2 may not be separable, and the data so far agree with this notion (Fig.S6). Repeller 1 is indeed independent of Repeller 2 plus the Attractor though, but given the passive property of Repeller 1 this seems almost trivial – why should impeded edge-cell crawling interfere significantly with convergent extension at the PS. Together, the results suggest that claiming modularity of DM units is not meaningful.

Overall, the emphasis of the manuscript is to view known avian gastrulation movements under a new aspect, but it may best advance our insight into morphogenetic movements if the authors discussed the relationships between Eulerian and Lagrangian representations of tissue flows more even-handedly and more critically. The weaknesses of the attractor/repeller concepts should be duly addressed together with its strengths. Practically, showing both approaches side by side in the figures could increase understanding. Respective Eulerian features could be shown on top of Figs. 3 and 4, as already done in Fig.2. As it stands, the one-sided promotion of the Lagrangian approach and the DM concept is not convincing in view of

the experimental and modeling results.

Minor points:

p.1: using "modular" in the title seems not warranted.

Fig.1B: The Attractor (as DOA) could in addition be depicted at same embryonic stage (HH1) as Repellers, for clarity of comparison.

p.4 last paragraph: "... even when Eulerian approaches struggle to capture meaningful patterns." Could that be more specific? Examples?

Fig.5B: why not show Attractor for "No mesoderm" and "Confined" here instead of in Fig.S6, and move "Twin", "Circular mesoderm" to Fig.S6?

Reviewer #2

(Remarks to the Author)

Synopsis:

The manuscript by Najera et al. addresses the relevant problem of directed tissue flows during embryonic development, using gastrulation in the chick embryo as model system.

The authors successfully match a careful quantification of tissue flows in 2D images and predictions of a mathematical model on a qualitative level in wildtype and several perturbation scenarios, building on their previous publication [39], but moving beyond by including additional perturbations.

Key points:

- The authors advocate a new concept, the "dynamic morphoskeleton", to characterize complex flow patterns in qualitative terms by a small set of flow attractors and repellers (first introduced in their ref. [37]).

This concept provides an elegant and concise description of flow patterns.

However, I am not fully convinced that it adds much to their mechanistic understanding, as the observed attractors and repellers are not the primary drivers of the flow, but emerge from the global dynamics of the tissue and the imposed boundary conditions (as is substantiated by the author's own mathematical model).

- The authors tested 2 hypotheses:

Repeller R1 results from opposition between embryo proper (EP) inward crawling and edge cell (EC) outward crawling (Figure 3); as an experimental test, the authors block epiboly by reducing adhesion to the Vitelline membrane (VM).

Repeller R2 results from elevated myosin activity and nematic alignment in the posterior part of embryo (Figure 4); as an experimental test, the authors reduce myosin activity by blocking mesendoderm induction.

Additionally, the authors show that both repellers R1 and R2 can be eliminated simultaneously by combining both perturbations (Fig. 5A, right column).

For completeness, the authors cite from their previous ref. [39] (cf. Figure 4 there) that the attractor results from the initial shape of the mesendoderm precursor and cell ingression (Figure 5B, which reproduces published data).

- The mathematical model describes the coupled dynamics of tissue flow, spatially varying myosin contractility, and nematic tissue polarity.

The model contains 7 unknown fit parameters, and depends on chosen initial and boundary conditions.

Although this slightly limits the quantitative predictive power of the model, the model is perfectly suited to rationalize observed flow patterns in qualitative terms in a robust manner.

Equations (1a) and (1c) generically describe fundamental physical processes (and are thus almost without alternative for the given level of model complexity, except maybe for the new and relevant active alignment term).

Equation (1b) makes specific assumptions on coarse-grained binding kinetics of myosin (already made in ref. [39]).

- The manuscript is mostly well written.

To increase readability, structure and logic could be highlighted more

(e.g., justifications for model assumptions are sometimes scattered throughout the manuscript and are often too brief).

Occasionally, a more neutral tone could increase readability.

Assessment:

This manuscript provides new data with high-quality analysis and comparison to theory, though maybe not a paradigm shift. In particular, the advancement relative to the previous publication ref. [39] warrants publication, though maybe in a more specialized journal.

Also, it is not fully clear how the suggested combinatorial control of attractors and repellers provides true mechanistic insight, as these attractors and repellers are not the primary drivers of tissue flows but an epiphenomenon of tissue-scale cell movements.

Comments:

1.

The authors should highlight the key advances relative to their previous work ref. [39] more clearly.

2.
The authors may want to comment on the model assumptions for their Eq. (1b), e.g., by showing that a simpler kinetics would fail to reproduce the observed flows. How would the predicted flow patterns change without mechano-sensitive myosin deactivation? How without tension propagation?
3.
Comparing the mathematical model from ref. [39] and that of the current manuscript, the main difference seems to be the additional active alignment and passive relaxation of the nematic order tensor. The authors should discuss the biological rationale to include these terms more extensively. How would flow patterns look like without these terms? (The only other difference in Eq. (1c) seem to be the use of either the alignment angle ϕ or the nematic tensor Q , which should be equivalent. The authors may want to confirm.)
4.
Can the authors combine the perturbations of attractor shape from ref. [39] with the new perturbations of repellers R1 and R2, to further substantiate their point of modular control of these high-level features of flow fields?

Minor comments:

- Why is a nonlinear passive relaxation $\sim s^2$ needed in Eq. (1b)? This term imposes $s \sim \sqrt{m}$ at steady-state - what is the rationale for this?
- The many acronyms sometime made reading a bit difficult.
- The terms Lagrangian and Eulerian Coherent Structures might have to be explained more for an interdisciplinary audience.
- The main text does not state that experiments are conducted in chicken (only the Supplemental Material does).
- Figure 5B (reproduced from ref. [39]) is barely discussed in the main text and essential information on the experimental design are missing.

Reviewer #3

(Remarks to the Author)

In this study, Serra, Weijer and colleagues address the presence and function of attractors and repellers determining the dynamic morphoskeleton in avian gastrulation. Previous work by the same groups has provided evidence for the presence of one attractor (the primitive streak) and two repellers. In the present work, the authors determine the specific function of the two repellers both theoretically (through development of a compressible active nematic model) and experimentally. I have a few major concerns that need to be addressed:

- 1/ The role of repeller 2 is experimentally only insufficiently addressed. Interfering with FGF signaling, and thus mesoderm induction, eliminates both repeller 2 and the attractor (PS). The authors should try to experimentally interfere with repeller 2 alone to test its specific function.
- 2/ It would also help explaining in more detail the relationship between the attractor and repeller 2 both theoretically and experimentally. Are there means to specifically address their functions and their functional interdependency?
- 3/ Some more insight into the cellular and mechanical mechanisms leading to the formation of repeller 1: can one interfere with the formation of cellular protrusions and/or contractility in EE cells? Are EP cells mechanosensitive adjusting their contractility to the pulling force by the EE cells? I assume some simple mechanical and biochemical; perturbation experiments could address this.

Collectively, this is a very nice study which would benefit from some more experimental work in order to clarify the specific functions especially of repeller 2.

Reviewer #4

(Remarks to the Author)

Version 1:

Reviewer comments:

Reviewer #1

(Remarks to the Author)

The authors made several changes to improve the manuscript, but unfortunately the main problem remains, the uncritical focus on the concept of a Dynamic Morphoskeleton (DM). Essential aspects of avian gastrulation are well known at the cellular level and can obviously be used (and are used by the authors) to model and predict overall tissue movements, and it is not convincingly shown why entities like Repellers and Attractors should be introduced. These terms, like also the expression “Morphoskeleton”, evoke the notions of substantial, causative agents while they denote abstractions with no causative potential or additional explanatory power beyond the known cell level mechanisms. This does not devalue the Lagrangian representation of tissue movements which can serve as heuristic tool and complement the Eulerian formalism. But expanding this aspect into a DM with modular Repellers and Attractors seems not only unnecessary but misleading. The authors themselves explain phenomena usually at the cell behavior level (cell crawling, intercalation, ingression, etc.) and Repellers/Attractors do not enter their model during its formulation. They are necessarily a predicted outcome if the model accurately predicts and describes gastrulation movements – but so are the Eulerian features. Overall, the DM is not critically evaluated.

Although the paper rightly draws attention to the Lagrangian representation of movements, contains some interesting new data, and supplies an improved mathematical model, with its focus on the DM it tends to re-formulate known features of chick gastrulation in a new terminology which, unfortunately, is not convincingly justified. Progress relative to previous work of the authors and others is otherwise limited, which suggests publication in a more specialized journal.

Additional specific points

Lines 15, 82, 285: epithelia are typically under tension and contract when cut free – to subsume this under “embryo size control” seems far-fetched.

Lines 41-43, red font: vague, not clear what is intended here.

Lines 76-84: modularity of Repellers: the weakness of this notion had been brought up in the previous comment of the reviewer (impossibility of separating Attractor/R2).

Line 157: “Immunostaining... confirms...”: the same immunostaining-based myosin densities were introduced in the previous paragraph as initial conditions in the model.

Lines 249-251: inserted text (red font) does not establish independence of Attractor from R2 (but independence of intercalation and ingression within the Attractor).

Lines 299-300: “sharp separation of cells ...”: not clear what this means.

Section S1.1 essentially justifies the Lagrangian vs. the Eulerian approach but not the DM concept. The limitations of the DM are addressed as essentially technical in nature (e.g. suitable choice of time frames) but not the important conceptual limitations addressed in the previous review and above.

Reviewer #2

(Remarks to the Author)

The manuscript by Najera et al. has improved substantially during revision, yet no new experiments were included. New experiments that e.g. combine the elimination of repellers (this study) with perturbations of the attractor (ref. [38]) would have considerably strengthened this study and allowed me to unanimously recommend it for publication in Nature Communications.

As it stands, I think publication in Nature Communications may still be possible (after addressing remaining points below) if there is a consensus that no new experiments should be done.

Remaining comments:

The revision was mainly restricted to a number of clarifications in the text.

No new experiments were included.

Figures 1, 2, 5 did not change; some additional model visualizations were included in Figs. 3 and 4.

Lagrangian and Eulerian frameworks are not well explained in main text (instead the reader is referred to the Supplementary Information).

The same applies to Coherent Structures (lines 41-42).

For the broad, interdisciplinary audience of Nature Communications, these terms should be introduced and explained in easy terms in the main text.

The description of the mathematical model (Eq. 1) improved and is now sufficient.

I like the discussion in terms of independent control of size and shape by eliminating either R1 or R2. To facilitate direct

comparison of shape, the authors should either quantify anisotropic EP shape, or at least show on overlay of WT and perturbed shapes. Is there a possibility to include information on EP-shape in Fig. 5A?

Line 105: I was surprised that the scalar order parameter 's' does not only characterize the alignment of actomyosin cables, but also their intensity. The authors should clarify the distinct roles played by the different scalars 's' and 'm'.

The physical meaning of 'm' is not clearly stated.

An interpretation wherein 'm' denotes local myosin concentration and 's' denotes local actin cable concentration is incompatible with Eq. (1a)

[force balance], where 'grad m' is said to describe isotropic active stress, which should depend on both myosin and actin.

This lack in clarity also becomes apparent in Fig. 2BC, where, if the above interpretation were true, myosin levels are high but actin levels are low in stage HH4.

The junior colleague with whom I am doing this review together noticed that different parameters were used in the mathematical model in this study (Table 1 in SI) as compared to ref. [39], Table 2 in SI. Also, a table mapping previous to new notation in the SI could be useful.

The authors should discuss the sensitivity of model predictions on model parameters in the SI.

Fig. 5B re-uses published data): this panel could either be eliminated, or a reference to refs. [38,39] should be added directly into the figure.

Although the manuscript still contains marketing language in some places, it has improved in this respect.

Minor comments

- Fig. 1 is very busy.

- Fig. 3 and 4 were expanded.

I like the parallel structure of Fig. 3 and 4.

Ideally, C and E and D and F could be placed next to each other to ease comparison (and a similar layout be used for Fig. 2F-I).

- Fig. 2C: I understand the colorcode denotes the product 'm*s'?

- Add labels EE, EP to Fig. 2A-E? Add labels A, R1, R2 to Fig. 5A? Add references to Figs. 2-4 in columns of Fig. 5?

- Line 73: explain 'gastrulation mode'?

Reviewer #3

(Remarks to the Author)

The revised manuscript has considerably improved over the previous version. All of my suggestions and criticisms have been adequately addressed. I now fully support publication of this study as is.

Reviewer #4

(Remarks to the Author)

Version 2:

Reviewer comments:

Reviewer #1

(Remarks to the Author)

The manuscript was thoughtfully revised. No objections to its publication remain.

Reviewer #2

(Remarks to the Author)

I appreciate the inclusion of new experiments (circular mesoderm after FGF treatment, Fig. S12), although elimination of repellents (this study) with perturbations of the attractor (ref. [38]) could have been even more insightful.

I would suggest that the authors include Fig. S12 (or at least a part of it) in Fig. 5 in the main text.

The explanation of the scalar order parameter 's' seems ok now.

The authors may want to add a sentence why the density of actin can be assumed spatially homogeneous.

All other previous comments are reasonably well addressed now.

Small points:

- In general, the manuscript improved in clarity.

To be more concise, the second sentence of the abstract seems void and could be deleted.

- line 38: "that summarize complex material trajectories": I did not understand this phrase.
- line 80: "single-cell behavior [46] modularity": I did not understand this phrase.
- line 80: "distributed cell behaviors": I did not understand this phrase.
- Swapping 3D-G and 3H (and 4D-G and 4h) to follow the visual logic of Figure 2 could have been nice.
- The image contrast of the dashed blue curves in Fig. 5B is not ideal, as the same colors are used in the color map.

Reviewer #4

(Remarks to the Author)

Reply to Reviewer 1

Reviewer's comments are in boldface, our responses are italicized and changes in the manuscript are in red.

The paper uses a refined version of an active matter model (Serra et al. 2023) to examine tissue flow in the avian embryo. It is based on the previously developed concept of a Dynamic Morphoskeleton (DM) (Serra et al. 2020) whose fundamental units are attractors and repellers of moving cells. As shown previously, in the avian embryo Repeller 1 separates extraembryonic tissue (EE) and the embryo proper (EP), Repeller 2 and an Attractor are located at the prospective mesoderm. In the present manuscript, experiments and modeling reveal the mechanistic origins of repellers, which suggests their independence as morphogenetic modules. The new model includes the EE and adds the modulation of anisotropic stress, and it correctly predicts gastrulation movements including the development of the EE-EP boundary.

1) The performance of the improved model captured in Equation 1 is impressive. However, instead of enforcing the DM concept the results obtained seem to weaken it. The DM had been introduced to replace the Eulerian representation of tissue flow through reference frame-dependent cell trajectories and streamlines by a frame-invariant description that identifies attractors and repellers, i.e. tissue regions that show maximal convergence or separation of cells over a given period of time. However, the issue of frame invariance seems not so important when the whole embryo is mapped anyway as in the present paper, instead of a select subregion. And it seems that mechanistic insights into gastrulation movements like convergent extension in the PS region or EE and EP moving apart in response to epiboly are easier to gain intuitively, at the cellular level (insets in Fig.1A), when streamlines are considered.

We appreciate the Reviewer's positive comment on the performance of our model. We failed to explain the use of the DM clearly. There are two main reasons we use the DM. i) It enables robust comparative analyses across embryos independent of embryo drifts and rotations without the need to carefully align them in time and space. As shown below, frame invariance remains important also when the whole embryo is imaged. For example, Fig. S2A (at 12h) shows an embryo drifting to the right, whose velocity snapshot and streamlines (Fig. 1 below at 12h) cannot reveal a sharp attractor marking the primitive streak (Fig. S2F right). We discuss these concepts in detail in [1]. ii) It reduces spatiotemporal velocities into discrete, interpretable units, which helped us discover new mechanisms (see later comments). We address the Reviewer's comment by modifying SI Sec. S1 and by adding a new SI Sec. S1.1, copied below.

The DM *i)* is invariant to time-dependent translations and rotations of the (arbitrary) coordinate system used to describe cell motion and global artifacts such as embryo drifts, **enabling robust comparative analyses, which would otherwise require careful spatiotemporal alignment of the embryos at each time point**; *ii)* is integrative, providing aggregate information along trajectories, guaranteeing robust results even with low-resolution wide-field microscopy; *iii)* **reduces spatiotemporal velocities into discrete, interpretable units.**

S1.1. DM and Eulerian quantities

We use chick gastrulation velocities (Fig. S2, [1]) to facilitate connections and differences between the DM and Eulerian quantities. 1) Velocity plots are unsuitable for locating regions of convergence and separation in time-dependent flows. For example, at the primitive streak (inside the red rectangle in Fig. S2A), the tissue is highly converging without revealing a clear signature in \mathbf{v} . 2) Instead, one should look at the frame invariant rate-of-strain tensor $\mathbf{S}(\mathbf{x}, t) = 1/2(\nabla\mathbf{v}(\mathbf{x}, t) + [\nabla\mathbf{v}(\mathbf{x}, t)]^T)$, (Fig. S2B-C) to locate local (in space and time) attraction or repulsion regions, as recently found [2, 3]. Specifically, denoting by $s_1(\mathbf{x}, t) \leq s_2(\mathbf{x}, t)$ the eigenvalues of $\mathbf{S}(\mathbf{x}, t)$, regions of low $s_1(\mathbf{x}, t)$ (high $s_2(\mathbf{x}, t)$) mark short time attractors (repellers), and the associated eigenvectors of $\mathbf{S}(\mathbf{x}, t)$ indicate the direction of convergence (separation).

Supplementary Figure S2: **Connections between DM and Eulerian quantities in Chick Gastrulation.** A) Chick Gastrulation Experimental velocities $\mathbf{v}(\mathbf{x}, t)$. t_0 corresponds to developmental stage HH1, and the red rectangle contains the Primitive Streak. B) $s_1(\mathbf{x}, t)$ field and directions (black bars) show the smallest eigenvalues (eigenvector) of the rate of strain tensor $\mathbf{S}(\mathbf{x}, t) = 1/2(\nabla\mathbf{v}(\mathbf{x}, t) + [\nabla\mathbf{v}(\mathbf{x}, t)]^T)$, showing the direction and intensity of maximal contraction rates. C) Same as B for maximal stretching rates and directions. D) Cell Trajectories. E) Deforming Lagrangian grid. F) Largest contraction ratio ${}_2\lambda_t^{t_0}(\mathbf{x}_t)$ along cell trajectories (Eq. S2) and contracting direction (white bar). High values of ${}_2\lambda_t^{t_0}(\mathbf{x}_t)$ mark Attractors. G) Largest separation ratio ${}_2\lambda_t^t(\mathbf{x}_0)$ along cell trajectories and expanding direction (white bar). High values of ${}_2\lambda_t^t(\mathbf{x}_0)$ mark Repellers.

Indeed, Fig. S2B shows high local convergence (or a short-time attractor [4]) at the Primitive Streak, where $\mathbf{v} \neq \mathbf{0}$ (Fig. S2A). This observation reveals that regions in deforming tissues can undergo large attraction/repulsion while

they move. By contrast, standard methods incorrectly look at frame-dependent saddle-type fixed points of the velocity (i.e., where $\mathbf{v} = \mathbf{0}$) to locate regions of the flows that undergo large convergence/separation, leading to incorrect predictions, as recently recognized [5]. 3) The (Eulerian) rate of strain tensor is frame invariant yet agnostic to cell paths. Tissue patches, instead, move and integrate deformations along their trajectories (Fig. S2D) as visualized by a deforming Lagrangian grid (Fig. S2E). The DM, made of attractors and repellers and computed from Eq. S2, precisely accounts for this cumulative deformation, revealing the intensity and direction of attraction and repulsion of tissue patches over a finite time (Fig. S2E-G). Comparing Fig. S2E-G with Fig. S2B-C shows how the DM reveals robust morphogenetic features that arise and shape over time, not contained in Eulerian quantities.

We recall this new SI material at the end of the third paragraph of the introduction, as copied below.

See Fig. S2 for Eulerian fields such as velocities and strain rates and their connection with the DM and other Lagrangian quantities.

Similarly, streamlines (i.e., trajectories of frozen velocity fields) plots are suboptimal to identify attractors and repellers, as shown below.

Figure 1: Streamlines for the velocity fields shown in Fig. S2A.

Please also see the reply to comment 6, highlighting when using streamlines and other Eulerian quantities instead of Lagrangian ones is appropriate.

2) On the other hand, the Lagrangian features extracted are kinematic units that are not directly linked to cell level mechanisms. For example, as shown here, Repeller 1 is generated by forces outside its domain, by epibolic edge-cell crawling right at the margin of the whole embryo and by general constriction in the EP – both features well represented in the Eulerian but not so obvious in the Lagrangian approach. There, the term Repeller evokes the idea of a local activity, not of a passive reaction to outside forces. One seems to be prompted to look for an active mechanism in the wrong place.

It is important to distinguish kinematic analysis of flows from the mechanisms (forces) generating them. In the former, it is meaningful to distinguish Eulerian vs Lagrangian quantities, as done in the previous comment. In the latter, it is not meaningful to make such distinctions as all forces and stresses are Eulerian (there is no force in our model involving Lagrangian quantities), consistent with the Reviewer’s comment. However, observing flow velocities or their DM (as in Fig. S2), neither reveal the underlying forces causing them, owing to their kinematic nature. We address the Reviewer’s comment by adding a paragraph at the end of Sec. S1.1, as copied below.

Kinematics vs Forces. Just as knowledge and inspection of Eulerian velocities do not reveal the underlying forces generating them (e.g. if cells moving with locally observed velocities are actively pulling or passively pushed), knowledge of attractors and repellers does not translate to knowledge of the underlying forces causing them or the presence of collocated active forces pulling towards and pushing away nearby tissues. Compressing

spatiotemporal velocities into three kinematic units (2 repellers and one attractor), however, guided our experiment design and force-based model development, enabling us to uncover their generating mechanisms and independent controllability.

In summary, consistent with the Reviewer’s comments, our model uses instantaneous (or Eulerian) forces. Yet model development and targeted perturbations were guided by the following questions: What mechanisms generate the two repellers? Can they be independently controlled? These questions wouldn’t have occurred to us by considering spatiotemporal velocities instead of discrete units. Please see also the reply to the next comment.

3) Moreover, as shown in the elegant confinement experiment (which could have been conceived without being aware of Repeller 1, just from curiosity about the effect of epiboly on gastrulation), the presence of Repeller 1 is of minor relevance for gastrulation.

We appreciate the Reviewer’s complimentary mention of our confinement experiment. In retrospect, it is easier to say what one may have tried out of curiosity, but in this case, our question was driven by the DM. First, we discovered that R2 could be eliminated with R1 intact, and then we asked the opposite. The combined results from experiments and simulations indicate that the intercalation and ingression of the mesoderm cells are not the only active drivers in chick gastrulation, but also suggest another independent, previously undocumented, active mechanism: EP myosin-driven apical constriction. While we found that the embryo gastrulates and forms proportionate axial structures without R1, the embryo becomes smaller. Size regulation and its consequential regulation of tissue patterning are important features of early embryogenesis [6, 7]. The implications of embryo size change for molecular patterning mechanisms critical at later stages is something we are following up on. We discuss the role of epiboly in evolution and the mechanisms embryos develop to control their size in Sec. S4.

4) As to Repeller 2, its origin can be traced to cell intercalation in the prospective mesoderm which also contributes to the Attractor (in addition to ingression). Thus, the same basic cellular process contributes to the Attractor and to the Repeller 2, which might be better seen when the Attractor is depicted as DOA at the same stage HH1 as Repeller 2 – but perhaps best when streamlines are considered. A conclusion from these latter considerations is that Attractor and Repeller 2 may not be separable, and the data so far agree with this notion (Fig.S6).

The Reviewer is right; Repeller 2 (R2) and the Attractor are closely linked—we failed to convey this clearly as this work focuses on repellers. However, streamlines and other Eulerian quantities do not clarify this connection, as they cannot resolve structures that develop over time as the tissue deforms, as shown in Fig. S2 and Fig. 1 above. Repeller 2 and the domain of attraction have been revealed by the DM [1]. Please see the reply to comment 6, highlighting when using streamlines and other Eulerian quantities instead of Lagrangian ones is appropriate. We have addressed the Reviewer’s comment by adding a sentence in the second paragraph of the section “Origin and Elimination of Repeller 2”, copied below.

These results show that while R1 and R2 are independently controllable, R2 is linked to the Attractor as mesoderm cells undergo active directed intercalation (contributing to R2 and Attractor) and ingression (contributing solely to the Attractor) (Sec. S3.2).

While separating R2 and the Attractor is unfeasible experimentally, we include a new supplementary section (S3.2 Coupling between R2 and the Attractor) exploring their theoretical separability. For a detailed explanation, please see our reply to comments 1 and 2 from Reviewer 3.

5) Repeller 1 is indeed independent of Repeller 2 plus the Attractor though, but given the passive property of Repeller 1 this seems almost trivial – why should impeded edge-cell crawling interfere significantly with convergent extension at the PS.

We have already partially addressed this comment in comment 3 above. Additionally, it is not trivial that R1 and convergent extension flows/PS formation are independent for the following reasons. i) Epibolic forces necessary to generate R1 induce tension that can travel fast over large distances in confluent epithelia, and could influence mechanosensitive myosin activity (driving gastrulation flows) and gene expression involved in mesoderm induction. ii) Historically, preventing epiboly by removing large EE areas led to poor development [8], which suggested an important role of epiboly in gastrulation flows. iii) It was not obvious that the active EP constriction would persist without mesoderm induction, allowing the generation of R1 in the absence of R2. This is a new important finding showing that chick embryos employ modular mechanisms to control size (affecting R1) and shape (affecting R2).

Together, the results suggest that claiming modularity of DM units is not meaningful.

We have addressed this comment in our previous replies. To summarize, the possibility of independently manipulating R1 and R2 led us to discover that chick embryos control size and shape with distinct mechanisms. In confined embryos, the size is significantly affected (Fig. 3G), but the shape change is not (Fig. 3E). Additionally, we do not claim modularity of R2 and the Attractor (see comment 4). To best reflect our results, we have modified the abstract to highlight the new mechanistic insights from our work, as copied below.

...Repeller 1, separating the embryo proper (EP) from extraembryonic (EE) tissues, **is associated with embryo size and arises from the tug-of-war between EE epiboly and EP isotropic myosin-induced active stress—which persists when mesoderm induction is blocked.** Repeller 2, bisecting the anterior and posterior PS and associated with embryo shape change, arises from anisotropic myosin-induced active intercalation in the mesendoderm. **Guided by our mathematical model that accounts for both embryonic and extraembryonic regions and predicts dynamic embryo geometry, we combined new confinement experiments with inhibition of mesendoderm induction and eliminated either one or both repellers, suggesting that chick embryos use distinct mechanisms to control their size and shape....**

6) Overall, the emphasis of the manuscript is to view known avian gastrulation movements under a new aspect, but it may best advance our insight into morphogenetic movements if the authors discussed the relationships between Eulerian and Lagrangian representations of tissue flows more even-handedly and more critically. The weaknesses of the attractor/repeller concepts should be duly addressed together with its strengths. Practically, showing both approaches side by side in the figures could increase understanding. Respective Eulerian features could be shown on top of Figs. 3 and 4, as already done in Fig.2. As it stands, the one-sided promotion of the Lagrangian approach and the DM concept is not convincing in view of the experimental and modeling results.

We have implemented the requests in the Reviewer's comment by making the following changes to the manuscript.

1. We have added Eulerian velocity fields in Figs. 3 and 4, copied below, and note that all Eulerian fields are present in every movie associated with the corresponding figures (Movies 1-4), as described in the figure captions, as well as Movies S1-S11.

Figure 2: **Repeller 1 Elimination.** Confining the embryo eliminates R1 but preserves R2. A,B,C,D) Model results solving Eq. 1 with $v_e = 0$ reflecting embryo confinement and all other parameters as in WT. A,B) Predicted model velocity and divergence at t_0 (A) and t_f (B). C) Deformed Lagrangian grid and final EP-EE boundary. D) Model repeller (R2), marked by a ridge of $2\lambda_{t_0}^{t_f}(x_0)$, normalized by the spatial maximum, and displayed on the initial undeformed configuration. E,F) Embryo confined in experiments by cauterizing the EP side of the vitelline membrane. E) Experimental Lagrangian grid overlaid on the bright-field image of a confined HH4 chick embryo, boundary located as in Fig. 2. F) Experiment repeller (R2), as in D. Compare C-F with the corresponding WT Figs. 2F-I. A,P labels mark the anterior-posterior axis. Movie 2 shows the time evolution of Lagrangian and Eulerian model fields as in Movie 1. G) Mean and standard deviation of experimental EP and EE areas over 15 hours ($N = 8$ per group). H) Doubly phosphorylated myosin light chain (cyan) and actin (phalloidin, red) in a representative section of a confined embryo.

Figure 3: **Repeller 2 Elimination.** Blocking mesendoderm induction eliminates R2 but preserves R1. **A,B** Predicted model velocity and divergence at t_0 (A) and t_f (B). C,D) Model results obtained solving Eq. (1) with uniform $m(\mathbf{x}_{EP}, t_0)$ reflecting the absence of mesendoderm. C) Deformed Lagrangian grid and final EP-EE boundary. D) Model repeller (R1), marked by a ridge of $2\lambda_{t_0}^{t_f}(\mathbf{x}_0)$, normalized by the spatial maximum, and displayed on the initial undeformed configuration. E,F) Mesendoderm induction blocked in experiments using $1\mu M$ LY2874455, a pan-FGF receptor inhibitor. E) Experimental grid overlaid on bright-field image of a treated HH4 chick embryo, boundary located as in Fig. 2. F) Experiment repeller (R2), as in D. Compare C-F with the corresponding WT Figs. 2F-I. Movie 3 shows the time evolution of model fields as in Movie 1. Movie attractor colorbar is normalized using the WT spatiotemporal maximum to emphasize the relative lack of deformation as in Fig. S7. G) Mean and standard deviation experimental EP and EE areas over 15 hours ($N = 4$ per group). H) Doubly phosphorylated myosin light chain (cyan) and actin (phalloidin, red) in a representative section of a treated embryo.

2. We have added a new SI section (S1.1) discussing the relationship between the DM and common Eulerian fields (see comment 1).
3. We have added a new paragraph in SI section S1.1, copied below, mentioning the limitations of the DM and when it is best to use streamlines or other Eulerian quantities.

Limitations of the DM. Contrary to Eulerian methods, quickly computable from snapshots of \mathbf{v} and straightforward to visualize, the DM and Lagrangian methods are more computationally demanding, requiring trajectory calculations. Additionally, owing to the finite-time integrative nature of the DM, care must be taken when choosing an appropriate time interval. If the time interval is too short, it may fail to resolve Lagrangian features that develop over time. If it is too long, it may fail to capture transient changes of \mathbf{v} over time. When the goal is detecting temporal changes in tissue flows [9] and velocity spatial patterns [10, 11], rather than their induced cumulative deformations, Eulerian snapshots of \mathbf{v} , streamlines, vorticity or short-time attractors and repellers [2, 3] are more appropriate.

Important: We note that the use of the DM vs Eulerian flow features was only a means to uncover the main results of our work, which deal with a mechanistic understanding of avian gastrulation flows. We have clarified these results in the text by modifying the last paragraph of the introduction, copied below.

This study, instead, addresses three open questions in avian gastrulation: the mechanistic origins of the morphogenetic repellers, their modularity, and the long-recognized embryo shape change from circle to pear-shaped [12, 13], missing a mechanistic explanation in existing models [14, 15, 16]. ...We found that *i*) active intercalation of the mesendoderm causes Repeller 2 and embryo shape change—the model predicts the characteristic EP shape change; *ii*) the competition between epiboly and myosin-induced isotropic constriction in the EP—which persists when mesoderm induction is blocked—causes Repeller 1 and contributes to embryo size control. *iii*) Using model-inspired perturbations and novel experiments to block epiboly, we found that *i*) and *ii*) are modular, suggesting that the embryo employs distinct mechanisms to control its shape and size.

Minor points:

p.1: using “modular” in the title seems not warranted.

We have removed modular from the title and updated it to: “Control of Tissue Flows and Embryo Geometry in Avian Gastrulation” to best reflect our main results.

Fig.1B: The Attractor (as DOA) could in addition be depicted at same embryonic stage (HH1) as Repellers, for clarity of comparison.

The domain of attraction is present in Fig. 1B bottom.

p.4 last paragraph: “... even when Eulerian approaches struggle to capture meaningful patterns.” Could that be more specific? Examples?

We have addressed the Reviewer’s comment by modifying the quoted paragraph, as copied below.

The DM remains tractable and identifies conserved features of the morphogenetic flows, even when they are not contained in Eulerian quantities [1, 17] (Sec. S1.1 and Fig. S2).

We have also provided a new supplementary Fig. S2 with examples (see comment 1). Additionally, in [17] we show that an undocumented repeller in zebrafish gastrulation flows, inaccessible from Eulerian fields, biases cell fate bifurcation of transcriptionally identical cells. Other examples include chick gastrulation flows at a later developmental stage [18]. Several other examples are available in the context of general fluid flows [2, 5, 19].

Fig.5B: why not show Attractor for “No mesoderm” and “Confined” here instead of in Fig.S6, and move “Twin”, “Circular mesoderm” to Fig.S6?

In the present work on controlling the repellers, the Attractor is always a thin streak and always co-occurs with R2. Instead, we opted to show the attractors from [14], which have different shapes to emphasize how this work extends and complements our previous work by controlling repellers in addition to attractors.

We thank the Reviewer for their extended and thoughtful comments, which improved our manuscript.

References

- [1] Mattia Serra, Sebastian Streichan, Manli Chuai, Cornelis J Weijer, and L Mahadevan. Dynamic morphoskeletons in development. *Proceedings of the National Academy of Sciences*, 117(21):11444–11449, 2020.
- [2] Mattia Serra and George Haller. Objective eulerian coherent structures. *Chaos: An Interdisciplinary Journal of Nonlinear Science*, 26(5), 2016.
- [3] Peter J Nolan, Mattia Serra, and Shane D Ross. Finite-time lyapunov exponents in the instantaneous limit and material transport. *Nonlinear Dynamics*, 100(4):3825–3852, 2020.
- [4] Carlo Sinigaglia, Francesco Braghin, and Mattia Serra. Optimal control of short-time attractors in active nematics. *Physical Review Letters*, 132(21):218302, 2024.
- [5] Mattia Serra, Pratik Sathe, Irina Rypina, Anthony Kirincich, Shane D Ross, Pierre Lermusiaux, Arthur Allen, Thomas Peacock, and George Haller. Search and rescue at sea aided by hidden flow structures. *Nature communications*, 11(1):2525, 2020.
- [6] Miloš Nikolić, Victoria Antonetti, Feng Liu, Gentian Muhaxheri, Mariela D Petkova, Martin Scheeler, Eric M Smith, William Bialek, and Thomas Gregor. Scale invariance in early embryonic development. *Proceedings of the National Academy of Sciences*, 121(46):e2403265121, 2024.
- [7] Dillan Saunders, Carlos Camacho-Macorra, and Benjamin Steventon. Spinal cord elongation enables proportional regulation of the zebrafish posterior body. *Development*, 152(1), 2025.
- [8] Ruth Bellairs, DR Bromham, and CC Wylie. The influence of the area opaca on the development of the young chick embryo. *Development*, 17(1):195–212, 1967.
- [9] Noah P Mitchell, Matthew F Lefebvre, Vishank Jain-Sharma, Nikolas Claussen, Marion K Raich, Hannah J Gustafson, Andreas R Bausch, and Sebastian J Streichan. Morphodynamic atlas for drosophila development. *bioRxiv*, pages 2022–05, 2022.
- [10] Rieko Asai, Shubham Sinha, Vivek N Prakash, and Takashi Mikawa. Cellular flows initiate left-right patterning prior to laterality gene expression in amniotes. *bioRxiv*, 2024.
- [11] Rieko Asai, Vivek N Prakash, Shubham Sinha, Manu Prakash, and Takashi Mikawa. Coupling and uncoupling of midline morphogenesis and cell flow in amniote gastrulation. *Elife*, 12:RP89948, 2024.
- [12] Nelson T Spratt Jr. Formation of the primitive streak in the explanted chick blastoderm marked with carbon particles. *Journal of Experimental Zoology*, 103(2):259–304, 1946.
- [13] Viktor Hamburger and Howard L Hamilton. A series of normal stages in the development of the chick embryo. *Journal of morphology*, 88(1):49–92, 1951.
- [14] Mattia Serra, Guillermo Serrano Nájera, Manli Chuai, Alex M Plum, Sreejith Santhosh, Vamsi Spandan, Cornelis J Weijer, and L Mahadevan. A mechanochemical model recapitulates distinct vertebrate gastrulation modes. *Science Advances*, 9(49):eadh8152, 2023.

- [15] Mehdi Saadaoui, Didier Rocancourt, Julian Roussel, Francis Corson, and Jerome Gros. A tensile ring drives tissue flows to shape the gastrulating amniote embryo. *Science*, 367(6476):453–458, 2020.
- [16] Aondoyima Ioratim-Uba, Tanniemola B Liverpool, and Silke Henkes. Mechanochemical active feedback generates convergence extension in epithelial tissue. *Physical Review Letters*, 131(23):238301, 2023.
- [17] Merlin Lange, Alejandro Granados, Shruthi VijayKumar, Jordão Bragantini, Sarah Ancheta, Yang-Joon Kim, Sreejith Santhosh, Michael Borja, Hirofumi Kobayashi, Erin McGeever, et al. A multimodal zebrafish developmental atlas reveals the state-transition dynamics of late-vertebrate pluripotent axial progenitors. *Cell*, 187(23):6742–6759, 2024.
- [18] Charlene Guillot, Yannis Djefal, Mattia Serra, and Olivier Pourquie. Control of epiblast cell fate by mechanical cues. *bioRxiv*, pages 2024–06, 2024.
- [19] George Haller. *Transport Barriers and Coherent Structures in Flow Data*. Cambridge University Press, 2023.

Reply to Reviewers 2, 4

Reviewer's comments are in boldface, our responses are italicized and changes in the manuscript are in red.

Synopsis: The manuscript by Najera et al. addresses the relevant problem of directed tissue flows during embryonic development, using gastrulation in the chick embryo as model system. The authors successfully match a careful quantification of tissue flows in 2D images and predictions of a mathematical model on a qualitative level in wildtype and several perturbation scenarios, building on their previous publication [39], but moving beyond by including additional perturbations.

Key points: - The authors advocate a new concept, the “dynamic morphoskeleton”, to characterize complex flow patterns in qualitative terms by a small set of flow attractors and repellers (first introduced in their ref. [37]). This concept provides an elegant and concise description of flow patterns. However, I am not fully convinced that it adds much to their mechanistic understanding, as the observed attractors and repellers are not the primary drivers of the flow, but emerge from the global dynamics of the tissue and the imposed boundary conditions (as is substantiated by the author's own mathematical model).

We appreciate the Reviewer's positive comments. The Reviewer's observation on the DM and other points are reiterated in the sections below, and we address them there.

- The authors tested 2 hypotheses: Repeller R1 results from opposition between embryo proper (EP) inward crawling and edge cell (EC) outward crawling (Figure 3); as an experimental test, the authors block epiboly by reducing adhesion to the Vitelline membrane (VM). Repeller R2 results from elevated myosin activity and nematic alignment in the posterior part of embryo (Figure 4); as an experimental test, the authors reduce myosin activity by blocking mesendoderm induction. Additionally, the authors show that both repellers R1 and R2 can be eliminated simultaneously by combining both perturbations (Fig. 5A, right column). For completeness, the authors cite from their previous ref. [39] (cf. Figure 4 there) that the attractor results from the initial shape of the mesendoderm precursor and cell ingression (Figure 5B, which reproduces published data).

- The mathematical model describes the coupled dynamics of tissue flow, spatially varying myosin contractility, and nematic tissue polarity. The model contains 7 unknown fit parameters, and depends on chosen initial and boundary conditions. Although this slightly limits the quantitative predictive power of the model, the model is perfectly suited to rationalize observed flow patterns in qualitative terms in a robust manner. Equations (1a) and (1c) generically describe fundamental physical processes (and are thus almost without alternative for the given level of model complexity, except maybe for the new and relevant active alignment term). Equation (1b) makes specific assumptions on coarse-grained binding kinetics of myosin (already made in ref. [39]).

- The manuscript is mostly well written. To increase readability, structure and logic could be

highlighted more (e.g., justifications for model assumptions are sometimes scattered throughout the manuscript and are often too brief). Occasionally, a more neutral tone could increase readability.

We clarify the logic of our study and elaborate on model assumptions in the replies to comments 1-3 below. We address the Reviewer's comment by removing words such as "remarkably" and similar throughout to maintain a more neutral tone.

Assessment: This manuscript provides new data with high-quality analysis and comparison to theory, though maybe not a paradigm shift. In particular, the advancement relative to the previous publication ref. [39] warrants publication, though maybe in a more specialized journal.

We appreciate the Reviewer's recognition of our manuscript's high-quality analysis and theoretical comparisons. We disagree with the lack of a paradigm shift as we have found important new results in avian gastrulation—which we may not have conveyed clearly—compared to the literature and ref. [39]. We summarized them below.

- 1. Developmental Biology: i) Our model and experiments suggest a new mechanism in avian gastrulation: myosin-driven apical constriction in the embryo proper, which contributes to embryo size control. Blocking mesoderm induction, myosin activity and the embryo's resistance to epiboly stretching is unaffected (Fig. 4). ii) Embryo shape change, instead, arises from active intercalation driven by myosin cables localized in the mesoderm—even without epiboly (Fig. 3). iii) Jointly, i) and ii) suggest that chick embryos adopt modular mechanisms affecting their size and shape.*
- 2. Modeling: Our model advances enabled the above findings. The model accounts 1) for the motion of both embryonic and extraembryonic regions; 2) captures distinct myosin dynamics in the extraembryonic and embryonic regions; and 3) introduces new terms for the creation and destruction of actomyosin cables. These features enabled us to predict force distributions that explain the embryo's area dynamics and shape change from circle to pear-shaped—a conspicuous feature [1, 2] missing a mechanistic explanation in all existing models [3, 4, 5].*
- 3. Experiments: We devised novel experiments to symmetrically block epiboly effects, overcoming the shortcomings of existing techniques (edge cells re-differentiate and regenerate asymmetric traction forces after standard cutting of edge cells).*

The points above are all new compared to ref. [39], and represent a significant improvement to the current mechanistic understanding of avian gastrulation. We thank the Reviewer for their comment and recognize that the above points did not emerge clearly in the current manuscript. To better highlight our main results, we modified the title and abstract of our paper, copied below. Other changes are included in the following comments.

Control of Tissue Flows and Embryo Geometry in Avian Gastrulation

...Repeller 1, separating the embryo proper (EP) from extraembryonic (EE) tissues, **is associated with embryo size** and arises from the tug-of-war between EE epiboly and EP isotropic myosin-induced active stress—which persists **when mesoderm induction is blocked**. Repeller 2, bisecting the anterior and posterior PS and associated with embryo shape change, arises from anisotropic myosin-induced active intercalation in the mesendoderm. **Guided by our mathematical model that accounts for both embryonic and extraembryonic regions and predicts dynamic embryo geometry**, we combined new confinement experiments with inhibition of mesendoderm induction and eliminated either one or both repellers, suggesting that chick embryos use distinct mechanisms to control their size and shape....

Also, it is not fully clear how the suggested combinatorial control of attractors and repellers provides true mechanistic insight, as these attractors and repellers are not the primary drivers of tissue flows but an epiphenomenon of tissue-scale cell movements.

We address the Reviewer’s comment by adding a paragraph at the end of SI Sec. S1.1., as copied below.

Kinematics vs Forces. Just as knowledge and inspection of Eulerian velocities do not reveal the underlying forces generating them (e.g. if cells moving with locally observed velocities are actively pulling or passively pushed), knowledge of attractors and repellers does not translate to knowledge of the underlying forces causing them or the presence of colocated active forces pulling towards and pushing away nearby tissues. Compressing spatiotemporal velocities into three kinematic units (2 repellers and one attractor), however, guided our experiment design and force-based model development, enabling us to uncover their generating mechanisms and independent controllability.

Specifically, model development and targeted perturbations were guided by the following questions: what mechanisms generate the two repellers? Can they be independently controlled? Addressing these questions led to new mechanistic understanding of avian gastrulation (see our reply to the “Assessment” comment above). These questions hadn’t occurred to us before knowing these discrete units.

Comments: 1. The authors should highlight the key advances relative to their previous work ref. [39] more clearly.

We address the Reviewer’s comments by modifying the last paragraph of the introduction, copied below.

This study, instead, addresses three open questions in avian gastrulation: the mechanistic origins of the morphogenetic repellers, their modularity, and the long-recognized embryo shape change from circle to pear-shaped [6, 2], missing a mechanistic explanation in existing models [3, 4, 5]. Our previous model [3] cannot address any of these questions because it enforces a fixed circular EP domain. To this end, here, we explicitly account for the EE region, the distinct EP-EE myosin dynamics, and new mechanisms to create and destroy aligned myosin cables. We found that *i*) active intercalation of the mesendoderm causes Repeller 2 and embryo shape change—the model predicts the characteristic EP shape change; *ii*) the competition between epiboly and myosin-induced isotropic constriction in the EP—which persists when mesoderm induction is blocked—causes Repeller 1 and contributes to embryo size control. *iii*) Using model-inspired perturbations and novel experiments to block epiboly, we found that *i*) and *ii*) are modular, suggesting that the embryo employs distinct mechanisms to control its shape and size.

2. The authors may want to comment on the model assumptions for their Eq. (1b), e.g., by showing that a simpler kinetics would fail to reproduce the observed flows. How would the predicted flow patterns change without mechano-sensitive myosin deactivation? How without tension propagation?

A) As noted in Sec. S3, sensitivity analyses of myosin kinetics were previously included in Sec. S4 of [3]. We address the Reviewer’s comment by adding a paragraph in SI Sec. S2.1 of the necessary features of our kinetic scheme and the experimental justification for the form we choose, copied below.

Our model requires an active stress instability to reproduce observed tissue flows and active stress patterns. Without this instability, gastrulation flows would cease (Table 3M). The instability arises from an unstable fixed point in the myosin kinetics (Fig. S3A), achieved in Eq. (S3) via negative feedback on the myosin’s detachment rate. Alternative kinetics, such as positive feedback on myosin’s recruitment could also suffice, but negative feedback on detachment is supported by experiments. In particular, myosin’s detachment rate exhibits mechanosensitivity, decreasing exponentially with tension [7], possibly due to a catch bond mechanism [8].

B) We clarify the reasoning behind model assumptions in paragraph 2 of the main text Mathematical Model section, copied below.

p_5 is the mechanosensitivity of the deactivation rate to cable tension [3], which results in an active stress instability and reflects the exponential decrease observed in single-molecule experiments [8, 7]. This active stress instability

ensures the divergent behavior of EP and EE regions, consistent with experiments (Fig. 2E, Sec. S2.1 and Fig. S3A).

C) We added a perturbation in Supplementary Table 3, copied below, to illustrate flow patterns without mechanosensitive deactivation.

Perturbation	Effect	Movie
L No Mechanosensitivity ($p_5 = 0$)	Eliminates R1, R2, Attractor	Movie S12

Last, because the tissue is fluid-like at gastrulation time scales, we previously argued that the advective transport of cells dominates the induction of myosin activity via tension propagation on individual cables, as explained in sections S1.3, S3, and S4.3 of [3].

3. Comparing the mathematical model from ref. [39] and that of the current manuscript, the main difference seems to be the additional active alignment and passive relaxation of the nematic order tensor. The authors should discuss the biological rationale to include these terms more extensively. How would flow patterns look like without these terms? (The only other difference in Eq. (1c) seem to be the use of either the alignment angle ϕ or the nematic tensor \mathbf{Q} , which should be equivalent. The authors may want to confirm.)

The dynamics of the nematic orientation ϕ and the nematic tensor $\mathbf{Q}(s, \phi)$ can be equivalent, but only when $s = 1$ (our earlier work implicitly imposed $s = 1$ everywhere at all times). This strong simplification, however, did not allow for the separate dynamics of isotropic and anisotropic active stresses, which have distinct spatiotemporal patterns (Fig. 1A, Fig. 2B-C, Movie 1). A key innovation of our new model is incorporating the dynamics for s (see Eq. S5b). When s is modeled, we found that advection and flow coupling alone cannot sustain convergent extension flows, requiring a compensatory mechanism ("active alignment" term in paragraph 2 of the main text "Mathematical Model"). Modeling \mathbf{Q} enables us to capture active cable formation (affecting s) as sufficient to recover Repeller 2, the Attractor and embryo shape change—inaccessible in our previous model. Flow patterns without these terms (Movie S4 and Fig. S3C-D) fail to reproduce observations. Section S2.4 rationalizes their inclusion in detail. We address the Reviewer’s comment by modifying the second paragraph of "Mathematical Model" to discuss the biological rationale more extensively, copied below.

This previous model, however, has important limitations... it assumes that all cables are perfectly aligned with the local average orientation ϕ ... To account for the differing degrees of cable alignment (i.e., the presence or absence and intensity of aligned actomyosin cables), we model a nematic order parameter s , which modulates anisotropic active stress $\propto m\mathbf{Q}$, where $\mathbf{Q} = s/2[\cos 2\phi, \sin 2\phi; \sin 2\phi, -\cos 2\phi]$ is the nematic tensor. The s dynamics, absent in the previous model, depends on flow coupling, active cable formation, and passive relaxation, enabling the creation, evolution and destruction of actomyosin cables (Fig. 1C-E, Sec. S2.2), as observed in experiments.

Notably, with only flow coupling and advection, convergent extension invariably destroys initial cable alignment, driving s to 0 in the vicinity of the streak (Fig. S3C, Sec. S2.2), inconsistent with experiments (Fig. 1 of [3]) and reflecting the loss of aligned high-myosin junctions from executing sequential cable contractions [9, 10]. To sustain convergent extension, an alignment mechanism is needed to counteract order destruction from self-induced flows (Fig. S3D). The active alignment term ($p_7 m\mathbf{Q}$) models actomyosin cable formation as an active process, conserving myosin but polarizing active stress to represent redistribution of myosin to junctions already under higher tension due to myosin activity [11, 12, 13, 14] (Fig. 1D). This agrees with experimental observations that myosin cables favor asymmetric stress propagation and that their formation can be inhibited by myosin inhibitors, as well as the hypothesis that they acquire their super-cellular alignment through a self-organizing stress-dependent mechanism [12]. It also reflects in the continuum setting a mechanism recently explored in vertex models, with

local tension feedback redistributing myosin to increase anisotropy while conserving m [10, 15, 16]. By contrast, passive relaxation ($-p_7 s^2 \mathbf{Q}$) reduces s accounting for cell shape rigidity and random intercalations [17, 10, 15] which would bring the tissue to an isotropic state (Fig. 1E). p_7 is t_c times the activity-induced alignment and passive relaxation rate (Sec. S2.5). Active alignment and passive relaxation terms, introduced in active nematics [18, 19], are yet unexplored in the context of morphogenesis.

4. Can the authors combine the perturbations of attractor shape from ref. [39] with the new perturbations of repellers R1 and R2, to further substantiate their point of modular control of these high-level features of flow fields?

We appreciate the Reviewer’s suggestion to combine the attractor shape perturbations from our previous work [3] with the new repeller elimination perturbations. However, this represents a major undertaking and is beyond the scope of this study, focused on repeller control and mechanisms to control embryo size and shape. Combining experimental attractor shape perturbations with those used here would require substantial time and resources. It would also require increasing model complexity accounting for a new variable that directly models ingression separate from apical constriction, unnecessary for the current work. To adequately address this last point, one should explicitly model (and measure) cell number density (ingression and division) and apical area dynamics, which is ongoing work (NSF PHY 2413073).

Minor comments: - Why is a nonlinear passive relaxation s^2 needed in Eq. (1b)? This term imposes $s \propto \sqrt{m}$ at steady-state - what is the rationale for this?

The quadratic relaxation term is consistent with nematic models (e.g. in [18, 20] from free energy considerations). However, even a simpler linear form does not change our results. We address the Reviewer’s comment by adding a new perturbation in Supplementary Table 3, referenced in section S2.2, copied below.

The quadratic form is typical in nematic models, but a linear form also suffices (Table 3M).

	Perturbation	Effect	Movie
M	Linear passive relaxation	Minimal effect	Movie S13

The steady-state dependence on m (discussed in section S2.2 and Fig. S3B), either in the quadratic or linear case, follows from our assumptions and also provides a natural initial condition $s(\mathbf{x}, t_0)$ (before motion starts, discussed in section S2.7) consistent with the initial presence of myosin cables in the posterior where myosin is higher and which relaxes when myosin is inhibited [12].

- The many acronyms sometime made reading a bit difficult.

We address the Reviewer’s comment by eliminating the following acronyms throughout to increase readability: particle image velocimetry (PIV), domain of attraction (DOA), vitelline membrane (VM) and edge cells (ECs).

- The terms Lagrangian and Eulerian Coherent Structures might have to be explained more for an interdisciplinary audience.

We address the Reviewer’s comment by adding a sentence in the second paragraph of the Introduction, copied below.

Eulerian Coherent Structures are dynamic curves in 2D and surfaces in 3D that organize material transport in the flow over short-time intervals, while Lagrangian Coherent Structures organize transport over finite-time intervals (Sec. S1).

We have also added a new SI Section S1.1 (not copied here) to facilitate connections between Lagrangian attractors and repellers, their Eulerian counterparts, and typical Eulerian quantities.

- The main text does not state that experiments are conducted in chicken (only the Supplemental Material does).

We address the Reviewer’s comment by specifying “chick embryos” in the Abstract, Results, and in figure captions, copied below.

Abstract: “suggesting that chick embryos use distinct mechanisms to control size and shape”

Results: “To test our model, we developed an imaging and analysis pipeline using bright-field microscopy, enabling us to simultaneously track tissue flows in up to ~ 16 chick embryos without fluorescent labels (Fig. S6).”

Figure 2: “Confocal image of a representative section of an HH4 stage WT chick embryo at the position indicated on the left.”

Figure 3: “Experimental Lagrangian grid overlaid on the bright-field image of a confined HH4 chick embryo”

Figure 4: “Experimental grid overlaid on bright-field image of a treated HH4 chick embryo”

Figure 5: “Repellers R1 and R2 can be controlled combinatorially by confining the chick embryo and inhibiting mesendoderm induction.”

- Figure 5B (reproduced from ref. [39]) is barely discussed in the main text and essential information on the experimental design are missing.

We address the Reviewer’s comment by expanding our description of Figure 5B in Discussion paragraph 2 and adding essential experimental details in the caption of Figure 5B, copied below.

Discussion: In our recent findings, we also controlled the shape of avian gastrulation’s Attractor (Fig. 1B) [3, 21], modifying its wild-type line shape (primitive streak) into a ring, dot, or thicker shorter line (Fig. 5B) by independently modulating the shape of the presumptive mesendoderm territory and the capacity of mesendoderm cells to undergo ingression. These modifications resemble specific aspects of gastrulation morphologies observed in amphibians, reptiles, and teleost fish, providing insights into the evolvability of gastrulation programs [22]. Modularity of attractors and repellers provide insights into the evolution of distinct aspects of development [23]—i.e., control of shape and size—and brings a new perspective to control natural and synthetic morphogenesis [24].

Caption: Control of attractor shape in model and experiments by modulating cells’ ability to ingress (16 hr treatment with 100 nM of VEGF receptor inhibitor axitinib) and initial mesendoderm shape (50 $\mu\text{g}/\text{ml}$ FGF2 on the hypoblast). See [21] for experimental details.

Reviewer #4 (Remarks to the Author): I co-reviewed this manuscript with one of the reviewers who provided the listed reports. This is part of the Nature Communications initiative to facilitate training in peer review and to provide appropriate recognition for Early Career Researchers who co-review manuscripts.

We thank Reviewer 4 and Reviewer 2 for their careful reading and thorough comments, which increased the clarity of our manuscript.

References

- [1] CO Whitman. A rare form of the blastoderm of the chick, and its bearing on the question of the formation of the vertebrate embryo. Journal of Cell Science, 2(91):376–398, 1883.
- [2] Viktor Hamburger and Howard L Hamilton. A series of normal stages in the development of the chick embryo. Journal of morphology, 88(1):49–92, 1951.

- [3] Mattia Serra, Guillermo Serrano Nájera, Manli Chuai, Alex M Plum, Sreejith Santhosh, Vamsi Spandan, Cornelis J Weijer, and L Mahadevan. A mechanochemical model recapitulates distinct vertebrate gastrulation modes. *Science Advances*, 9(49):eadh8152, 2023.
- [4] Mehdi Saadaoui, Didier Rocancourt, Julian Roussel, Francis Corson, and Jerome Gros. A tensile ring drives tissue flows to shape the gastrulating amniote embryo. *Science*, 367(6476):453–458, 2020.
- [5] Aondoyima Ioratim-Uba, Tanniemola B Liverpool, and Silke Henkes. Mechanochemical active feedback generates convergence extension in epithelial tissue. *Physical Review Letters*, 131(23):238301, 2023.
- [6] Nelson T Spratt Jr. Formation of the primitive streak in the explanted chick blastoderm marked with carbon particles. *Journal of Experimental Zoology*, 103(2):259–304, 1946.
- [7] Melanie F Norstrom, Philip A Smithback, and Ronald S Rock. Unconventional processive mechanics of non-muscle myosin iib. *Journal of Biological Chemistry*, 285(34):26326–26334, 2010.
- [8] Claudia Veigel, Justin E Molloy, Stephan Schmitz, and John Kendrick-Jones. Load-dependent kinetics of force production by smooth muscle myosin measured with optical tweezers. *Nature cell biology*, 5(11):980–986, 2003.
- [9] Emil Rozbicki, Manli Chuai, Antti I Karjalainen, Feifei Song, Helen M Sang, René Martin, Hans-Joachim Knölker, Michael P MacDonald, and Cornelis J Weijer. Myosin-ii-mediated cell shape changes and cell intercalation contribute to primitive streak formation. *Nature cell biology*, 17(4):397–408, 2015.
- [10] Fridtjof Brauns, Nikolas H Claussen, Eric F Wieschaus, and Boris I Shraiman. The geometric basis of epithelial convergent extension. *eLife*, 13, 2024.
- [11] Maria Duda, Natalie J Kirkland, Nargess Khalilgharibi, Melda Tozluoglu, Alice C Yuen, Nicolas Carpi, Anna Bove, Matthieu Piel, Guillaume Charras, Buzz Baum, et al. Polarization of myosin ii refines tissue material properties to buffer mechanical stress. *Developmental cell*, 48(2):245–260, 2019.
- [12] E. Rozbicki, M. Chuai, A.I. Karjalainen, F. Song, H.M. Sang, R. Martin, H. Knölker, M.P. MacDonald, and C.J Weijer. Myosin-II-mediated cell shape changes and cell intercalation contribute to primitive streak formation. *Nat Cell Biol*, 17(4):397, 2015.
- [13] Valentina Ferro, Manli Chuai, David McGloin, and Cornelis J Weijer. Measurement of junctional tension in epithelial cells at the onset of primitive streak formation in the chick embryo via non-destructive optical manipulation. *Development*, 147(3):dev175109, 2020.
- [14] Rodrigo Fernandez-Gonzalez, Sérgio de Matos Simoes, Jens-Christian Röper, Suzanne Eaton, and Jennifer A Zallen. Myosin ii dynamics are regulated by tension in intercalating cells. *Developmental cell*, 17(5):736–743, 2009.
- [15] Nikolas H Claussen, Fridtjof Brauns, and Boris I Shraiman. A geometric-tension-dynamics model of epithelial convergent extension. *Proceedings of the National Academy of Sciences*, 121(40):e2321928121, 2024.
- [16] Rastko Sknepnek, Ilyas Djafer-Cherif, Manli Chuai, Cornelis Weijer, and Silke Henkes. Generating active t1 transitions through mechanochemical feedback. *Elife*, 12:e79862, 2023.
- [17] Scott Curran, Charlotte Strandkvist, Jasper Bathmann, Marc de Gennes, Alexandre Kabla, Guillaume Salbreux, and Buzz Baum. Myosin ii controls junction fluctuations to guide epithelial tissue ordering. *Developmental cell*, 43(4):480–492, 2017.

- [18] L Giomi, L Mahadevan, B Chakraborty, and MF Hagan. Banding, excitability and chaos in active nematic suspensions. Nonlinearity, 25(8):2245, 2012.
- [19] Aphrodite Ahmadi, M Cristina Marchetti, and Tanniemola B Liverpool. Hydrodynamics of isotropic and liquid crystalline active polymer solutions. Physical Review E, 74(6):061913, 2006.
- [20] Pierre-Gilles De Gennes and Jacques Prost. The physics of liquid crystals. Number 83. Oxford university press, 1993.
- [21] Manli Chuai, Guillermo Serrano Nájera, Mattia Serra, Lakshminarayanan Mahadevan, and Cornelis J Weijer. Reconstruction of distinct vertebrate gastrulation modes via modulation of key cell behaviors in the chick embryo. Science Advances, 9(1):eabn5429, 2023.
- [22] Guillermo Nájera Serrano and Cornelis J. Weijer. The evolution of gastrulation morphologies. Development, 150(7):dev200885, 04 2023.
- [23] Christian Peter Klingenberg. Morphological integration and developmental modularity. Annual review of ecology, evolution, and systematics, 39:115–132, 2008.
- [24] Carlo Sinigaglia, Francesco Braghin, and Mattia Serra. Optimal control of short-time attractors in active nematics. Phys. Rev. Lett., 132:218302, May 2024.

Reply to Reviewer 3

Reviewer's comments are in boldface, our responses are italicized and changes in the manuscript are in red.

In this study, Serra, Weijer and colleagues address the presence and function of attractors and repellers determining the dynamic morphoskeleton in avian gastrulation. Previous work by the same groups has provided evidence for the presence of one attractor (the primitive streak) and two repellers. In the present work, the authors determine the specific function of the two repellers both theoretically (through development of a compressible active nematic model) and experimentally. I have a few major concerns that need to be addressed:

1/ The role of repeller 2 is experimentally only insufficiently addressed. Interfering with FGF signaling, and thus mesoderm induction, eliminates both repeller 2 and the attractor (PS). The authors should try to experimentally interfere with repeller 2 alone to test its specific function.

The Reviewer raises an important point about R2 and its relation to the attractor. Our model suggests (see also 2/) that to experimentally interfere with R2 alone would require changes in i) patterning: The mesendoderm should be induced in-situ along the midline, and ii) in cell behaviours: The mesendoderm cells should be only able to ingress, instead of ingressing and medially intercalating. Currently, it is not possible to experimentally reproduce these conditions in a living embryo, since we cannot induce the mesendoderm directly along the midline, or prevent cell intercalation without affecting cell ingression (note that myosin inhibition blocks both processes simultaneously). However, we use our model to elucidate this interesting point. We address the Reviewer's comment by modifying the second paragraph of the main text section "Origin and elimination of Repeller 2" (A) and the third paragraph of the Discussion (B). Please see also the reply to the next comment.

(A) These results show that while R1 and R2 are independently controllable, R2 is linked to the Attractor as mesendoderm cells undergo active directed intercalation (contributing to R2 and Attractor) and ingression (contributing solely to the Attractor) (Sec. S3.2).

(B) In contrast, R2 and the Attractor jointly reflect embryo shape changes and primitive streak formation from an initially crescent-shaped mesendoderm.

2/ It would also help explaining in more detail the relationship between the attractor and repeller 2 both theoretically and experimentally. Are there means to specifically address their functions and their functional interdependency?

We address the Reviewer's comment by adding a new SI Sec. S3.2, referenced in the additions above (see 1/), and a new model perturbation in SI Table 3, separating R2 and the Attractor, both copied below.

S3.2 Connection Between R2 and the Attractor

R1 and R2 arise from distinct mechanisms and can be experimentally separated. In contrast, R2 and the Attractor

are inseparable in chick, as they are generated by convergent extension and ingression of the same mesendoderm tissue region. From purely kinematic considerations, convergent-extension flows necessarily create an attractor (from convergence) and a repeller (from extension). For example, the simplest convergent-extension flow velocity $\mathbf{v} = [-x, y]$ generates an attractor (y-axis) and repeller (x-axis). In the mesendoderm, additional isotropic convergence associated with cell ingression sharpens the Attractor. Consistent with this argument, inhibiting ingression (Fig. 2J in [1]) using the vascular endothelial growth factor (VEGF) receptor inhibitor axitinib preserves convergent extension motion, but reduces isotropic convergence (Fig. 2K-L in [1]), generating a thicker (i.e., less sharp) attractor and a smaller domain of attraction (Fig. 4C-D in [2]).

Our model predicts that convergent extension arises from anisotropic active forces (generating an attractor, repeller pair). Simultaneously, isotropic active forces associated with ingressions sharpen the attractor. For these reasons, initializing myosin directly along the midline instead of as a crescent converging to the midline, the model—without anisotropic active forces—generates a line attractor without R2 (Table 3N). This scenario resembles the early mouse embryo, where mesendoderm is pre-patterned along the midline [3], and cells undergo apical contraction and ingression without oriented intercalation. This theoretical demonstration of attractor-repeller separability suggests additional modular potential and offers insights into how different vertebrate species may have evolved distinct gastrulation modes (with or without a repeller) by modifying the relative contributions of isotropic and anisotropic active forces. The presence of R2 in chick may help support compartmentalization [4] by reducing communication between nearby cells separated into the anterior and posterior primitive streak.

Perturbation	Effect	Movie
N Line initial condition, no anisotropic myosin activity ($s = 0$)	Attractor without R2	Movie S14

3) Some more insight into the cellular and mechanical mechanisms leading to the formation of repeller 1: can one interfere with the formation of cellular protrusions and/or contractility in EE cells?

We appreciate the Reviewer’s question. While alternative methods exist to impede epiboly movements, each comes with significant limitations. We discuss these in detail in paragraph 2 of the main text, section “Origin and elimination of Repeller 1”. Techniques that disrupt cellular protrusions would typically cause detachment from the vitelline membrane, which is crucial for maintaining tension and minimizing out-of-plane motion. Similarly, interfering with embryo contractility generally disrupts cable and streak formation (related to R2), though such perturbations are possible in the model (see supplementary Table 3A-C). Our confinement technique was developed as a precise, less disruptive approach to interfere with epiboly. We have validated that confined embryos remain healthy well past primitive streak formation (Fig. S10) and believe our new technique best balances restricting epiboly with preserving other processes and maintaining tissue integrity.

Are EP cells mechanosensitive adjusting their contractility to the pulling force by the EE cells? I assume some simple mechanical and biochemical perturbation experiments could address this.

This is an excellent point that we are starting to address in separate ongoing work (NSF PHY 2413073), which is beyond the scope of this current work. To properly address this question, one needs 1) a viscoelastic model, where the elastic part enables the tissue to bear external tension. 2) Explicitly model (and measure) the cell apical area dynamics—which further depend on division and ingression rates—to properly account for cytoskeletal remodeling and myosin response to tension. This is required, especially in the chick embryo, where the dynamics of the apical area in the embryo proper and extraembryonic tissues are markedly different [5]. This task requires extensive

quantification of cell area dynamics, division and ingression and will lead to a more complicated model that is unnecessary for the scope of this work. Additionally, it would require local and global active stretching experiments, which are under way but will constitute a large independent study.

On the other hand, in the experiments presented here, blocking epiboly does not change the level of myosin activity in obvious ways (see Fig. 2E, Fig. S9). These confined embryos are smaller after 2 days of development, but proportional (Fig. S10). These results suggest the presence of regulatory mechanisms as well as longer-term developmental implications. Properly addressing this regulatory question deserves a dedicated investigation. We address the Reviewer's comment by modifying the fourth Discussion paragraph, copied below.

In future work, we plan to... *ii) incorporate divisions, ingressions and viscoelastic tissue rheology to investigate cell density dynamics and integration of mechanical signals propagated by epibolic pulling;...*

Collectively, this is a very nice study which would benefit from some more experimental work in order to clarify the specific functions especially of repeller 2.

We appreciate the Reviewer's positive feedback on our study. As a final note, we refer the Reviewer to our recent preprint [4], revealing the theoretical implications of repellers for cell-cell signaling via morphogen diffusion during morphogenetic movements. This work shows that repellers not only help understand morphogenesis but also reveal how morphogenesis mediates morphogen patterning in dynamic tissues. Specifically, it suggests a potential function for R2 in restricting communication between cells forming the anterior and posterior primitive streak (Fig. 4 in [4]). Investigating the function of R2 in patterning processes presents an exciting future direction, requiring dynamic modeling of diffusible morphogens (e.g., Nodal, FGF, Wnt, etc.) along with new measurements and perturbation experiments. This, however, lies beyond this study's scope: the mechanical origin and modularity of repellers. To address the Reviewer's comment, we modify the third Discussion paragraph (A) and add a final item to our planned future work (B), copied below.

(A) *In chick, sharp separation of cells anterior and posterior to R2 may aid in compartmentalization along the PS [4].*

(B) *iv) clarify the function of each repeller in simultaneous patterning dynamics. We recently proposed that morphogenetic repellers constrain cell-cell communication via diffusible morphogens to support compartmentalization [4], here between EE and EP (R1) and between cells forming anterior and posterior PS (R2).*

We thank the Reviewer for their thoughtful questions, which enhanced the quality and clarity of our manuscript.

References

- [1] Manli Chuai, Guillermo Serrano Nájera, Mattia Serra, Lakshminarayanan Mahadevan, and Cornelis J Weijer. Reconstruction of distinct vertebrate gastrulation modes via modulation of key cell behaviors in the chick embryo. *Science Advances*, 9(1):eabn5429, 2023.
- [2] Mattia Serra, Guillermo Serrano Nájera, Manli Chuai, Alex M Plum, Sreejith Santhosh, Vamsi Spandan, Cornelis J Weijer, and L Mahadevan. A mechanochemical model recapitulates distinct vertebrate gastrulation modes. *Science Advances*, 9(49):eadh8152, 2023.
- [3] Margot Williams, Carol Burdsal, Ammasi Periasamy, Mark Lewandoski, and Ann Sutherland. Mouse primitive streak forms in situ by initiation of epithelial to mesenchymal transition without migration of a cell population. *Developmental Dynamics*, 241(2):270–283, 2012.
- [4] Alex M Plum and Mattia Serra. Morphogen patterning in dynamic tissues. *bioRxiv*, pages 2025–01, 2025.

- [5] Guillermo Serrano Nájera. *Analysis and modulation of cell behaviours driving avian gastrulation*. PhD thesis, School of Life Sciences, University of Dundee, 2021.

Reply to Reviewer 1

Reviewer's comments are in boldface, our responses are italicized and changes in the manuscript are in red.

1. The authors made several changes to improve the manuscript, but unfortunately the main problem remains, the uncritical focus on the concept of a Dynamic Morphoskeleton (DM). Essential aspects of avian gastrulation are well known at the cellular level and can obviously be used (and are used by the authors) to model and predict overall tissue movements, and it is not convincingly shown why entities like Repellers and Attractors should be introduced. These terms, like also the expression "Morphoskeleton", evoke the notions of substantial, causative agents while they denote abstractions with no causative potential or additional explanatory power beyond the known cell level mechanisms.

In our previous reply to comment 2, we clarified that the Dynamic Morphoskeleton (attractors and repellers) are kinematic and not causative. To address this aspect of the Reviewer's comment, we clarify this promptly in the introduction (copied below) and carefully remove any language with a causative connotation throughout.

Page 1: ...These are attractors and repellers in the dynamical systems sense, not to be confused with causative factors like chemoattractants.

We highlight that the main contribution of this paper is on the mechanistic understanding of how avian embryos control their size and shape (comment 3 below), not the Dynamic Morphoskeleton, established in prior work [1] and used across model systems [2-4].

2. This does not devalue the Lagrangian representation of tissue movements which can serve as heuristic tool and complement the Eulerian formalism. But expanding this aspect into a DM with modular Repellers and Attractors seems not only unnecessary but misleading. The authors themselves explain phenomena usually at the cell behavior level (cell crawling, intercalation, ingression, etc.) and Repellers/Attractors do not enter their model during its formulation. They are necessarily a predicted outcome if the model accurately predicts and describes gastrulation movements – but so are the Eulerian features. Overall, the DM is not critically evaluated.

Here, the DM was a useful tool providing a robust cumulative description of the observed kinematics, especially pinpointing evolving boundaries between regions of different behaviors, which are inherently Lagrangian and not contained in Eulerian quantities [1, 5] (Fig. S2). This kinematic quantification converts complex spatiotemporal flows over 15 hours into discrete units, which guided our experiments/model development. Specifically, we focused on the two repellers and asked i) which mechanisms generate them; and ii) whether those mechanisms can be modulated independently. These targeted questions enabled us to find new mechanistic insights about avian gastrulation, summarized in comment 3 below. We critically evaluate the DM, highlighting Eulerian approaches to study avian gastrulation, how Lagrangian methods complement Eulerian ones, and give each of their limitations.

Discussion: Previous studies... ...instantaneous flow patterns.

SI Sec. 1.1 Limitations of the DM: ...In general, there is... ...remains a challenging task.

Finally, some of the Reviewer's suggested limitations reflect standard features of attractors and repellers in dy-

namical systems. By analogy, studies of gene regulatory networks frequently consider attractors (stable cell types) and repellers (decision boundaries) summarizing gene expression trajectories without entering the models [6–10].

3. Although the paper rightly draws attention to the Lagrangian representation of movements, contains some interesting new data, and supplies an improved mathematical model, with its focus on the DM it tends to re-formulate known features of chick gastrulation in a new terminology which, unfortunately, is not convincingly justified. Progress relative to previous work of the authors and others is otherwise limited, which suggests publication in a more specialized journal.

Our current manuscript’s focus on the DM may have obscured our main results on mechanisms underlying embryo geometry control, which are new to existing literature. To this end, we make several changes to the text.

Introduction: This study, instead, investigates the origin of dynamic EP geometry... ...distributed cell behaviors.

Some of these results have been appreciated by the Reviewer, reporting the performance of our model “impressive” (comment 1) and the new confinement experiments “elegant” (comment 3) in the previous review round.

We also rewrote the abstract: Embryonic tissues undergo coordinated flows during avian gastrulation to establish the body plan. How cell behaviors collectively orchestrate these movements to sculpt the embryo’s dynamic geometry remains poorly understood. Here, we elucidate how the interplay between embryonic and extraembryonic tissues affects the chick embryo’s size and shape. These two distinct geometric changes are each associated with dynamic curves across which trajectories separate (kinematic repellers). Through physical modeling and experimental manipulations of both embryonic and extraembryonic tissues, we selectively eliminate either or both repellers in model and experiments, revealing their mechanistic origins. We find that embryo size is affected by the competition between extraembryonic epiboly and embryonic myosin-driven contraction—which persists when mesoderm induction is blocked. Instead, the characteristic shape change from circular to pear-shaped arises from myosin-driven cell intercalations in the mesendoderm, irrespective of epiboly. These findings elucidate modular mechanisms controlling avian gastrulation flows and provide a mechanistic basis for the independent control of embryo size and shape during development.

Finally, we removed any unnecessary text focusing on the DM throughout.

Additional specific points

4. Lines 15, 82, 285: epithelia are typically under tension and contract when cut free – to subsume this under “embryo size control” seems far-fetched.

We state that the EP and EE are under tension. Here, however, we show that EP active contractility limits EP expansion relative to the EE (see deformation in Fig. 2F-G; and Fig. 3C), even without PS formation (Fig. 4D-E; Fig. 4C). Without this active contractility, the EP would expand as the EE region (Table 3B-C), inconsistent with data. ‘Embryo size control’ refers to this intrinsic EP mechanisms that influence its size, as demonstrated. We address the Reviewer’s comment by modifying the “Origin and elimination of Repeller 1” section, as copied below.

Page 5: ...Epiboly contributes to global tension [11–13] that propagates to the EP, as severing the EP-EE boundary causes both regions to contract [12]. Yet, R1 shows that EP and EE cells separate, implying distinct expansion rates. Tension can arise from passive stretching and active contractility. We hypothesize that the EP resists epiboly-driven expansion with isotropic myosin activity (Figs. 1A, 2A,B,E), which constricts cells’ cortical cytoskeletons (active contractility), allowing them to bear the isotropic tension contributed by epiboly without stretching like EE cells.

5. Lines 41-43, red font: vague, not clear what is intended here.

We address the Reviewer’s comment by adding a clear explanation (suggested also by Reviewer 2/4). Page 1: **To this end... ...‘repellers’ (Fig. S1).**

6. Lines 76-84: modularity of Repellers: the weakness of this notion had been brought up in the previous comment of the Reviewer (impossibility of separating Attractor/R2).

We claim that the Repellers have modular properties (via their independent mechanistic origins) and show it in the model and experiments (Figs. 2-5). We do not claim that R2 and the Attractor are modular and explained their connection in Sec. S3.2 (comment 4, previous round). We address the Reviewer’s comment by remarking on their connection in paragraph 3 of the “Origin and elimination of Repeller 2” section, copied below.

Mesoderm is necessary for both R2 and the Attractor, co-occurring in chick because the crescent-shaped mesoderm’s elongation into the PS requires convergent extension, contributing to both R2 and a linear attractor (Sec. S3.2).

This dependence in no way weakens the notion that R1 and R2 have modular properties. Please see also the new FGF experiments (Fig. S12 in our reply to Reviewers 2/4), showing a (circular) attractor without R2.

7. Line 157: “Immunostaining... confirms...”: the same immunostaining-based myosin densities were introduced in the previous paragraph as initial conditions in the model.

The model’s initial condition is based on immunostaining data in [2]. We clarify this by modifying the last paragraph before the Results section, as copied below.

For the initial condition of Eq. (1b), we use immunostaining data indicating minimal myosin activity in the interior EE throughout gastrulation, while myosin activity increases over time in the EP (Figs. 1E, S10 in [2]), consistent with an m instability (Fig. S3A, Sec. S2.1).

8. Lines 249-251: inserted text (red font) does not establish independence of Attractor from R2 (but independence of intercalation and ingression within the Attractor).

See reply to comment 6 above. We revise the referenced lines for clarity: Consistent with model results, the embryo retains a circular geometry (Fig. 4E, Fig. S13), keeping R1, which becomes circularly symmetric, while eliminating both R2 and the Attractor (Figs. 4G, S7C). These results suggest that R1 arises from distinct mechanisms, independent of those jointly generating R2 and the Attractor.

9. Lines 299-300: “sharp separation of cells ...”: not clear what this means.

We address the Reviewer’s comment by clarifying this sentence: In chick, initially close cells starting on opposite sides of R2 experience large separation during gastrulation (for visualization, see Fig. 3A of [1]),...

10. Section S1.1 essentially justifies the Lagrangian vs. the Eulerian approach but not the DM concept. The limitations of the DM are addressed as essentially technical in nature (e.g. suitable choice of time frames) but not the important conceptual limitations addressed in the previous review and above.

We have addressed the Reviewer’s comment in our reply to comment 2 above.

Overall, we emphasize that the present work is about the mechanistic insights we gained into the dynamics of chick embryo geometry (size and shape) using biophysical modeling and experiments. Following the Reviewer’s comments, we have made substantial changes throughout, minimizing emphasis on the DM. These changes best reflect our results. We thank the Reviewer for their comments, which enhanced the clarity of our manuscript.

References

- [1] Mattia Serra, Sebastian Streichan, Manli Chuai, Cornelis J Weijer, and L Mahadevan. Dynamic morphoskeletons in development. *Proceedings of the National Academy of Sciences*, 117(21):11444–11449, 2020.
- [2] Mattia Serra, Guillermo Serrano Nájera, Manli Chuai, Alex M Plum, Sreejith Santhosh, Vamsi Spandan, Cornelis J Weijer, and L Mahadevan. A mechanochemical model recapitulates distinct vertebrate gastrulation modes. *Science Advances*, 9(49):eadh8152, 2023.
- [3] Merlin Lange, Alejandro Granados, Shruthi VijayKumar, Jordão Bragantini, Sarah Ancheta, Yang-Joon Kim, Sreejith Santhosh, Michael Borja, Hirofumi Kobayashi, Erin McGeever, et al. A multimodal zebrafish developmental atlas reveals the state-transition dynamics of late-vertebrate pluripotent axial progenitors. *Cell*, 187(23):6742–6759, 2024.
- [4] Charlene Guillot, Yannis Djeflal, Mattia Serra, and Olivier Pourquie. Control of epiblast cell fate by mechanical cues. *bioRxiv*, pages 2024–06, 2024.
- [5] G. Haller. Lagrangian coherent structures. *Annual Rev. Fluid. Mech*, 47:137–162, 2015.
- [6] Stuart A Kauffman. The origins of order: Self-organization and selection in evolution. In *Spin glasses and biology*, pages 61–100. World Scientific, 1992.
- [7] Sui Huang, Gabriel Eichler, Yaneer Bar-Yam, and Donald E Ingber. Cell fates as high-dimensional attractor states of a complex gene regulatory network. *Physical review letters*, 94(12):128701, 2005.
- [8] Néstor Saiz, Laura Mora-Bitria, Shahadat Rahman, Hannah George, Jeremy P Herder, Jordi Garcia-Ojalvo, and Anna-Katerina Hadjantonakis. Growth-factor-mediated coupling between lineage size and cell fate choice underlies robustness of mammalian development. *Elife*, 9:e56079, 2020.
- [9] David A Rand, Archishman Raju, Meritxell Sáez, Francis Corson, and Eric D Siggia. Geometry of gene regulatory dynamics. *Proceedings of the National Academy of Sciences*, 118(38):e2109729118, 2021.
- [10] Meritxell Sáez, Robert Blassberg, Elena Camacho-Aguilar, Eric D Siggia, David A Rand, and James Briscoe. Statistically derived geometrical landscapes capture principles of decision-making dynamics during cell fate transitions. *Cell systems*, 13(1):12–28, 2022.
- [11] DAT New. The adhesive properties and expansion of the chick blastoderm. *Development*, 7(2):146–164, 1959.
- [12] Ruth Bellairs, DR Bromham, and CC Wylie. The influence of the area opaca on the development of the young chick embryo. *Development*, 17(1):195–212, 1967.
- [13] JR Downie. The mechanism of chick blastoderm expansion. *Development*, 35(3):559–575, 1976.

Reply to Reviewers 2, 4

Reviewer’s comments are in boldface, our responses are italicized and changes in the manuscript are in red.

The manuscript by Najera et al. has improved substantially during revision, yet no new experiments were included. New experiments that e.g. combine the elimination of repellers (this study) with perturbations of the attractor (ref. [38]) would have considerably strengthened this study and allowed me to unanimously recommend it for publication in Nature Communications. As it stands, I think publication in Nature Communications may still be possible (after addressing remaining points below) if there is a consensus that no new experiments should be done.

We appreciate the Reviewer’s positive note and address their comment by conducting two new experiments/simulations requested by this Reviewer only: (1) The circular mesoderm (FGF) attractor perturbation described in [38-39], now also imaging and modeling the EE tissue and (2) combining the circular mesoderm perturbation with confinement. We report our new results in Fig. S12 and main text ‘Origin and Elimination of Repeller 2’, copied below.

Supplementary Figure S12: Circular Mesoderm. Repellers, attractors and Lagrangian deformation with circular mesoderm (FGF treatment) in unconfined (A) and confined (B) conditions. Colorbars for repellers (attractors) mark $2\lambda_{t_0}^{t_f}(\mathbf{x}_0)$ ($2\lambda_{t_f}^{t_0}(\mathbf{x}_f)$). Circular mesoderm is modeled by replacing the crescent-shaped myosin initial condition with a radially symmetric Gaussian function with radius 0.25, standard deviation 0.03, and amplitude $A_m = 0.15$.

Mesoderm is necessary for both R2 and the Attractor, co-occurring in chick because the crescent-shaped mesoderm’s elongation into the PS requires convergent extension, contributing to both R2 and a linear attractor (Sec. S3.2). To test the role of mesoderm geometry, we used FGF to generate a circular mesoderm domain [1, 2]. In the model and experiments, this circular geometry results in convergence towards a circular attractor [3], instead of an extending line attractor, reinforcing a symmetric R1 and eliminating R2 (Fig. S12A). Furthermore, we show that combining FGF addition with confinement, the circular attractor persists, R2 remains absent, and circular contraction recovers a weak R1 (Fig. S12B). This clarifies the role of mesoderm geometry in jointly generating R2 and the (line) Attractor, associated with embryo shape change.

Other combined perturbations involving the modulation of cell ingression are beyond the current model's scope, as described in response to comment 4 in the previous round of review.

Remaining comments:

The revision was mainly restricted to a number of clarifications in the text. No new experiments were included. Figures 1, 2, 5 did not change; some additional model visualizations were included in Figs. 3 and 4. Lagrangian and Eulerian frameworks are not well explained in main text (instead the reader is referred to the Supplementary Information). The same applies to Coherent Structures (lines 41-42). For the broad, interdisciplinary audience of Nature Communications, these terms should be introduced and explained in easy terms in the main text.

We address the Reviewer's comment by limiting discussion of Eulerian Coherent Structures to the SI (these are used in the main text) and adding simple explanations of Lagrangian Coherent Structures in the introduction.

Page 1: **To this end... 'repellers'** (Fig. S1).

The description of the mathematical model (Eq. 1) improved and is now sufficient.

I like the discussion in terms of independent control of size and shape by eliminating either R1 or R2. To facilitate direct comparison of shape, the authors should either quantify anisotropic EP shape, or at least show on overlay of WT and perturbed shapes. Is there a possibility to include information on EP-shape in Fig. 5A?

We thank the Reviewer for the positive note and address their comment by quantifying the EP shapes. Fig. 5A depicts repellers, which are scalar fields over the reference configuration \mathbf{x}_0 , whereas the deformed EP shape is on deformed configuration \mathbf{x}_t . We report EP shape changes in a new Fig. S13, copied below.

Supplementary Figure S13: **Shape quantification.** EP's circularity ($4\pi(\text{area})/(\text{perimeter})^2$) in different treatments. Black squares mark EP circularity in model simulations. Control (Fig. 2), Confined (Fig. 3), LY (Fig. 4), LY + confined (Fig. S7D). LY2874455 $1\mu\text{M}$, $N = 8, 8, 4, 4$, respectively.

We reference this figure in the main text sections describing the corresponding case, as copied below.

Page 4. *Wild-Type*: See Movie 1 for the time evolution of these fields and Fig. S13 for shape quantification.

Page 7. *R1 Elimination*: ...the embryo still becomes pear-shaped (Fig. 3E, Fig. S13)...

Page 8. *R2 Elimination*: ...the embryo retains a circular geometry (Fig. 4E, Fig. S13)...

Page 8. *Combined elimination*: ...resulting in no shape change (Fig. S13),...

Line 105: I was surprised that the scalar order parameter 's' does not only characterize the alignment

of actomyosin cables, but also their intensity. The authors should clarify the distinct roles played by the different scalars ‘s’ and ‘m’. The physical meaning of ‘m’ is not clearly stated. An interpretation wherein ‘m’ denotes local myosin concentration and ‘s’ denotes local actin cable concentration is incompatible with Eq. (1a) [force balance], where ‘grad m’ is said to describe isotropic active stress, which should depend on both myosin and actin. This lack in clarity also becomes apparent in Fig. 2BC, where, if the above interpretation were true, myosin levels are high but actin levels are low in stage HH4.

The reviewer’s intuition is correct: the order parameter ‘s’ characterizes myosin anisotropy, and does not separately model actin. Scalar values in Fig. 2C (ms) are lower than 2B (m) because $s \in [0, 1]$. We clarify this in the text by modifying the paragraph before Eq. 1, copied below.

To account for the differing degrees of cable alignment (i.e., the presence or absence ~~and intensity~~ of aligned actomyosin cables), we model a nematic order parameter s , which modulates anisotropic active stress $\propto m\mathbf{Q}$, where $\mathbf{Q} = s/2[\cos 2\phi, \sin 2\phi; \sin 2\phi, -\cos 2\phi]$ is the nematic tensor ~~and m is the local fraction of available myosin generating active stress~~ (Sec. S2.1).

The junior colleague with whom I am doing this review together noticed that different parameters were used in the mathematical model in this study (Table 1 in SI) as compared to ref. [39], Table 2 in SI. Also, a table mapping previous to new notation in the SI could be useful.

We address the Reviewer’s comment by adding a new SI table mapping the old and new parameters, referenced at the beginning of the SI Mathematical Model section.

SI Page 4: See Table 4 for a map between old [39] and new parameters. Table 4 is in SI page 17.

The model is robust to broad changes in the parameter values, as discussed in the next comment and SI Section 3.

The authors should discuss the sensitivity of model predictions on model parameters in the SI.

In SI Section 3 and Table 3, we provide sensitivity analysis to parameters p_2 , p_5 and other model perturbations. We address the Reviewer’s comment by extending SI Table 3 (rows O to X) with sensitivity analyses to the remaining parameters and a discussion in SI Section 3.

SI Page 10: Increasing.... ...yields similar results. Additional to SI Table 3 are on page 11.

Fig. 5B re-uses published data): this panel could either be eliminated, or a reference to refs. [38,39] should be added directly into the figure.

We address the Reviewer’s comment by referencing [39] directly in Fig. 5B. All 5 revised main text figures are included in the revised manuscript.

Although the manuscript still contains marketing language in some places, it has improved in this respect.

To increase readability throughout, we have minimized text overly focusing on the DM or previous work and highlighted instead new insights. We also removed all languages such as ‘fundamental,’ ‘essential,’ and similar.

Minor comments

- Fig. 1 is very busy.

We have simplified Fig. 1, removing most of the in-panel text.

- **Fig. 3 and 4** were expanded. I like the parallel structure of **Fig. 3 and 4**. Ideally, **C and E and D and F** could be placed next to each other to ease comparison (and a similar layout be used for **Fig. 2F-I**).

We follow the Reviewer's suggestion by placing similar panels side-by-side in Figs. 2-4.

- **Fig. 2C**: I understand the colorcode denotes the product 'm*s'?

The Reviewer is correct. We modify the Fig. 2 caption for clarity: B,C) t_f distribution of isotropic (B, color bar for m) and anisotropic (C, color bar for ms with directors as in A) myosin activity.

- Add labels **EE, EP** to **Fig. 2A-E**? Add labels **A, R1, R2** to **Fig. 5A**? Add references to **Figs. 2-4** in columns of **Fig. 5**?

We follow the Reviewer's suggestions and modify Figs. 2 and 5 accordingly.

- **Line 73**: explain 'gastrulation mode'?

We address the Reviewer's comment, copied here: ...vertebrate gastrulation modes, i.e., their tissue flows, attractor geometries and internalization mechanisms...

We thank the Reviewers for carefully reading our manuscript and their constructive comments.

References

- [1] Cantas Alev, Yuping Wu, Yukiko Nakaya, and Guojun Sheng. Decoupling of amniote gastrulation and streak formation reveals a morphogenetic unity in vertebrate mesoderm induction. *Development*, 140(13):2691–2696, 2013.
- [2] Manli Chuai, Guillermo Serrano Nájera, Mattia Serra, Lakshminarayanan Mahadevan, and Cornelis J Weijer. Reconstruction of distinct vertebrate gastrulation modes via modulation of key cell behaviors in the chick embryo. *Science Advances*, 9(1):eabn5429, 2023.
- [3] Mattia Serra, Guillermo Serrano Nájera, Manli Chuai, Alex M Plum, Sreejith Santhosh, Vamsi Spandan, Cornelis J Weijer, and L Mahadevan. A mechanochemical model recapitulates distinct vertebrate gastrulation modes. *Science Advances*, 9(49):eadh8152, 2023.